# Exploring access to health and social supports for intimate partner violence (IPV) survivors during stressful life events (SLEs)—A scoping review

**Dina Idriss-Wheeler**[1], **Xaand Bancroft**[1], **Saredo Bouraleh**[2], **Marie Buy**[1], **Sanni Yaya**[3‡], **Ziad El-Khatib**[4,5‡]*

**1** Interdisciplinary School of Health Sciences, University of Ottawa, Ottawa, Canada, **2** School of Epidemiology and Public Health, Faculty of Medicine, University of Ottawa, Ottawa, ON, Canada, **3** The George Institute for Global Health, Imperial College London, London, United Kingdom, **4** Department of Global Public Health, Karolinska Institutet, Stockholm, Sweden, **5** University of Global Health Equity (UGHE), Kigali, Rwanda

‡ ZEK and SY are co-senior authors.
* ziad.el-khatib@ki.se

## Abstract

### Background

Survivors of intimate partner violence (IPV) often face increased incidents of violence during stressful life events (SLEs) such as economic recessions, environmental disasters, and pandemics. These events can diminish the effectiveness of both formal (e.g., health, social, justice, labor, community) and informal (e.g., friends, family, neighbors) support systems. Additionally, SLEs exacerbate existing health and social inequities, making it necessary to understand the accessibility of support services during these times. This scoping review investigates access to services by individuals experiencing IPV during SLEs in high-income countries.

### Approach

A comprehensive search was conducted across several electronic databases including MEDLINE (OVID), Embase (OVID), PsychInfo (OVID), CINAHL (EBSCO), Global Health (EBSCO), Gender Watch (ProQuest), Web of Science, and Applied Social Sciences Index & Abstracts (ProQuest), along with the search engine Google Scholar. This search, which imposed no date restrictions, was extended through May 22nd, 2024. Key search terms were developed from prior literature and in consultation with an expert librarian, focusing on 'stressful life events,' 'intimate partner violence,' and 'access to services.'. Each study was screened and extracted by two reviewers and conflicts were resolved through discussion or a third reviewer.

### Results

The search across eight databases and citation searching resulted in a total of 7396 potentially relevant articles. After removing 1968 duplicates and screening 5428 based on titles

**Data Availability Statement:** All relevant data are within the paper and its Supporting Information files.

**Funding:** The author(s) received no specific funding for this work.

**Competing interests:** The authors have declared that no competing interests exits.

and abstracts, 200 articles underwent full abstract review. Ultimately, 74 articles satisfied the inclusion criteria and were selected for further analysis. The analysis focused on barriers and facilitators to access, identifying challenges within Survivors' support systems, redirected resources during crises, and complex control dynamics and marginalization. Over 90% of the literature included covered the recent COVID-19 pandemic. Addressing these challenges requires innovative strategies, sustained funding, and targeted interventions for high-risk subgroups.

## Conclusion

This scoping review systematically outlined the challenges and enabling factors influencing the availability of support services for Survivors of IPV during SLEs. It underscores the need for robust, culturally sensitive health and social support mechanisms, and policies. Such measures are essential to better protect and assist IPV Survivors and their service providers during these critical times. Furthermore, it is imperative to integrate the insights and expertise of the violence against women (VAW) sector into emergency planning and policy-making to ensure comprehensive and effective responses that address the unique needs of Survivors in crises.

## Introduction

Intimate partner violence (IPV) refers to behaviors within an intimate relationship that cause physical, sexual, psychological or financial harm, and acts of controlling or coercive behaviors by a current or former partner (including spouse, boyfriend/girlfriend, dating/sexual partner) [1–3]. IPV Survivors often face serious physical and mental health effects, leading to significant health inequities [4]. IPV is associated with a range of mental health issues such as depression, anxiety disorders, sleep disorders, self-harm, and post-traumatic stress disorder (PTSD) [5]. Survivors of IPV are at higher risk of physical health injuries, chronic pain, and reproductive health issues, as well as heightened risk of brain injury which can exacerbate mental health issues in the long term [5, 6]. Furthermore, the trauma associated with IPV can lead to a diminished sense of self-worth and self-esteem, which is not only linked to adverse health outcomes but also to negative health behaviors such as smoking and substance use [7, 8].

Carlson (2014) describes stressful life events (SLEs) as undesirable, unscheduled, nonnormative, and uncontrollable discrete, observable events with a generally clear onset and offset that usually signify major life changes [9]. The adverse effects of IPV are significantly exacerbated during events such as public health emergencies (e.g., pandemics, epidemics), natural disasters (e.g., floods, earthquakes, hurricanes, fires), and economic recessions, making it important to examine the experiences of IPV during such stressful life events (SLEs) (10–14)." [10–14]. These crises often amplify pre-existing tensions within relationships, potentially escalating abusive behaviors and increasing the vulnerability and health inequities of IPV Survivors. Public health emergencies (e.g., the COVID-19 pandemic) and natural disasters (e.g., Hurricane Katrina) can disrupt essential services, including access to shelters, counseling, and legal resources [15]; thus, leaving IPV Survivors with fewer options for escape or assistance. Economic recessions (e.g., the 2008 Economic recession) can contribute to the onset or escalation of IPV in households, resulting in intensified financial stressors and interpersonal conflicts [13].

Survivors of IPV facing the added challenges of SLEs rely on a combination of formal and informal supports for their health and well-being [16, 17]. Formal supports encompass a network of services provided by professionals and organizations, including shelters, healthcare, hotlines, community-based programs, legal aid, social welfare, housing, criminal justice services, and counseling. These resources ensure immediate safety, legal protection, financial support, and emotional and physical recovery for Survivors during times of crisis. Formal supports tend to offer a structured pathway out of abusive situations. In tandem, informal supports from friends, family members, and community networks provide a vital lifeline for IPV Survivors [18, 19]. These informal supports offer emotional comfort, practical assistance, and a sense of belonging which can be particularly invaluable during times of heightened stress [18, 20]. The combined strength of both formal and informal supports creates a resilient safety net for Survivors which is threatened during SLEs.

A scoping review of literature can reveal key concepts, contributing to the understanding of social and health support service accessibility during disasters and emergencies. This research topic is important, especially in light of increased climate-related disasters and economic recessions [21, 22] as well as the aftermath of the COVID-19 pandemic, which offers an opportunity to update and complement the existing literature on impacts of the pandemic on IPV. By examining IPV experiences and unique challenges within the context of SLEs, service providers and policymakers can create programs and policies that address the key issues associated with IPV, inform emergency response plans, and ensure that resources and support are readily available during and in the aftermath of SLEs.The full rationale for this scoping review is detailed in the published scoping review protocol [23].

This scoping review examined, summarized and characterized existing literature on informal and formal social and health supports accessed by Survivors of IPV during SLEs, including pandemics, natural disasters, and economic recessions in high-income countries. The aim was to understand key factors and issues involved in access to formal and informal supports during SLEs by Survivors of IPV to inform work or research related to improving access to services during these types of events. This review is specifically focused on high-income countries to inform a comprehensive study aimed at assessing access to both informal (including family, friends, and neighbors) and formal support mechanisms for individuals who experienced IPV during the COVID-19 lockdowns in Ontario, Canada. This review intentionally selects countries with comparable social and health infrastructure frameworks to facilitate a realistic comparative analysis of support service accessibilities.

## Methods

The scoping review was guided by the following research question: W*hat is known about access to services for individuals who experience IPV during SLEs in high-income countries*? More specifically, the review explores the following sub-questions:

■ What are the barriers and facilitators to accessing and providing informal and formal supports by Survivors of IPV during SLEs?

■ What approaches are used for sub-groups at higher risk of isolation, such as gender diverse individuals, women, girls, minority/ethnic racialized populations, and individuals living in rural remote regions, to access support during SLEs?

■ What are the key lessons learned from attempts to access or provide formal and informal supports to Survivors of IPV during SLEs?

## Protocol and registration

The methodological frameworks for scoping reviews as described by Arksey and O'Malley (2005) [24] and the Joanna Briggs Institute (2020) [25] were used. The Preferred Reporting Items for Systematic Reviews and Meta-Analysis extension for Scoping Reviews (PRISMA-ScR) as described by Tricco et.al. (2018) guided the organization of the review [26]. The review is registered in the Open Science Framework (doi:10.17605/OSF.IO/RE724). Further information and details of the study approach are available in the protocol, which has undergone peer review and publication [23].

## Eligibility criteria

The scoping review was focused on examining how Survivors of IPV accessed necessary supports during significant life events within high-income countries. It included perspectives from both survivors and service providers to offer a comprehensive overview of the support landscape. The inclusion and exclusion criteria were methodically structured around the PICOT framework as follows:

Population:

■ The review targeted individuals aged 13 or older who had experienced IPV, defined as any behavior within an intimate relationship that causes physical, psychological, or sexual harm to those involved. The rationale for including the 13+ age range in this review stems from evidence that the experience of IPV can begin as early as age 13 [2]. Additionally, during times of disasters, such as pandemics and natural emergencies, vulnerable populations— including women, children, and gender diverse persons—face a heightened risk of violence, underscoring the need to include younger adolescents in the review to capture the full scope of IPV experiences [27, 28].

Phenomenon of Interest/Context:

■ The primary phenomenon of interest was the impact of stressful life events (SLEs) on these individuals. SLEs encompassed scenarios such as pandemics (specifically COVID-19), natural disasters or environmental emergencies, and economic recessions.

■ The geographic context was confined to high-income countries, as classified by the World Bank and similar economic indices, to ensure the findings were relevant to nations with similar economic and social structures.

Outcomes:

■ The primary outcome investigated was the accessibility to and experience of accessing/providing both formal and informal supports for individuals who had experienced IPV during SLEs. Formal supports included organized services from institutions like violence against women shelters, healthcare facilities, social service organizations, community service groups, spiritual/religious bodies, labor sectors, and criminal justice systems. Informal supports encompassed assistance provided by personal connections, such as family, friends, and neighbors.

Types of Evidence Sources:

• The review included studies directly involving IPV survivors and articles that captured the perspectives of service providers assisting these individuals. This encompassed qualitative, quantitative, and mixed-methods research.

Exclusion Criteria:

■ Studies were excluded if they focused on populations other than those specified, addressed IPV in contexts other than SLEs, or were conducted in low- or middle-income countries.

■ Articles addressing domestic violence focusing on the child-parent/parent-child dynamic versus IPV between intimate partner relationships.

■ Studies that only focused on prevalence, incidence, associations and likelihood of IPV in SLEs.

■ Excluded document types included literature reviews, systematic reviews, scoping reviews, commentaries, opinion pieces, editorials, and conference abstracts.

Publications in languages other than English and French were also excluded, with no restrictions on the year of publication.

## Information sources and search

The following electronic databases were searched for peer-reviewed studies: MEDLINE (OVID), Embase (OVID), PsycInfo (OVID), CINAHL (EBSCO), Global Health (EBSCO), Gender Watch (ProQuest), Web of Science, Applied Social Sciences Index & Abstracts (Pro-Quest), and Google Scholar. The selection of the eight databases, and Google Scholar search engine, was guided by a consultation with a population health science librarian, ensuring a comprehensive and relevant literature search aligned with the scope of the study. These databases were deemed most appropriate for topics within public health and social sciences. Key terms and medical subject headings (MeSH) were based on previous literature and consultation with the expert librarian. The major concepts included 'stressful life events' AND 'intimate partner violence' AND 'access to services'. In the original study protocol, it was planned to include grey literature to broaden the scope of the review. However, during the course of the study, an overwhelming volume of peer-reviewed articles that fulfilled the inclusion criteria was encountered. Consequently, a strategic decision was made to focus exclusively on these articles, leading to a deviation from the initial protocol. This adjustment was deemed necessary to maintain manageability and ensure a thorough analysis of the available peer-reviewed literature. Supplement 1 (S1 File) outlines the search strategy used in Medline which was created in consultation with the University of Ottawa Population Health librarian and modified from previously completed search strategies using similar key terms [29–31]. The Medline search strategy was appropriately translated to the other databases listed above. There were no date restrictions on the search which includes articles up to May 22nd, 2024.

## Selection of sources of evidence

Search results from the databases were imported into the web-based platform Covidence [32]. Titles and abstracts were screened by two reviewers and conflicts resolved through discussion or a third reviewer. The principal investigator (DIW) screened all articles. Among the co-screeners, XB reviewed approximately 40% of the articles (~2172), while MB and SB each reviewed about 30% (~1628) to ensure that each article was screened by two researchers. The same method of four reviewers was used during the full-text screening, any conflicts or disagreements were resolved through discussion and, if necessary, the involvement of a third reviewer. The studies that met the inclusion criteria outlined in the eligibility criteria were compiled into a final list for further analysis. A PRISMA flow diagram was automatically generated through Covidence and a modified Page et al. (2021) PRISMA diagram created (Fig 1) [33].

## Data charting process and data items

The team used the Covidence automated extraction tool to extract the data. The principal investigator (DIW) created an extraction tool (S2 File) that guided the charting process, specifying the elements to be extracted for each included article [34]. This tool—modified from JBI Manual for Evidence Synthesis Data Charting tool for Scoping Reviews—was piloted by the researchers using three studies, and the extraction parameters were subsequently adjusted accordingly. The team of reviewers (DIW, XB, SB, & MB) continued to meet regularly throughout the extraction phase to ensure consistency when extracting article elements and that the team followed the research question(s). Similar to the screening phases, DIW extracted all 74 articles, while XB, SB, and MB shared the extraction of the 74 articles to ensure each article was extracted by at least two researchers. In Covidence, any discrepancies between extractions require a resolution to determine the final consensus on extracted material. The two extractors engaged in discussions when necessary to resolve these conflicts effectively. Once consensus was reached, the article underwent complete extraction and proceeded to the synthesis and analysis phase.

## Synthesis of results

Extracted data were exported from Covidence to Microsoft Excel allowing for analysis of qualitative and quantitative data (S2 Table). Simple frequency counts of the key concepts, populations, and results or outcomes were quantitatively analyzed were relevant. The extracted qualitative data were analyzed using Braun and Clark's (2021) thematic analysis approach [35] and informed by Mak and Thomas's (2022) guidelines for conducting thematic analysis in scoping reviews [36]. The data relevant to the research question and sub-questions were iteratively coded and analyzed, leading to the creation of themes based on emerging patterns across the codes (S3 Table). The focus was on key findings related to barriers and facilitators to accessing formal (health and social services) and informal (family, friends, neighbours) supports by individuals who experience IPV during SLEs. Additionally, implications of findings and recommendations based on lessons learned, such as the knowledge gained from the process or experience of providing services or trying to access services during SLEs were summarized.

## Results

### Selection of sources of evidence

Initially, 7396 records were identified from nine databases. After removing 1968 duplicates, 5428 records underwent title and abstract screening. Subsequently, 5228 records were excluded during this phase. A further detailed review of the remaining 200 studies identified 74 articles that met the inclusion criteria. An additional two articles were found through citation screening. This resulted in a final list of 74 papers that were analyzed in this review; refer to the PRISMA Flowchart (Fig 1) for details on the selection process of sources of evidence. Supplement 3 (S1 Table) contains three tables: Table A details the 5,228 articles excluded at the title and abstract screening stage with reasons for exclusion, Table B lists the 126 articles excluded during the full-text screening stage with corresponding reasons, and Table C outlines the 74 articles included in the scoping review.

### Characteristics of sources of evidence

Table 1 presents a detailed summary of the characteristics of the included articles. The articles included in this review were published between 1999–2024. Over half of the studies, 57%

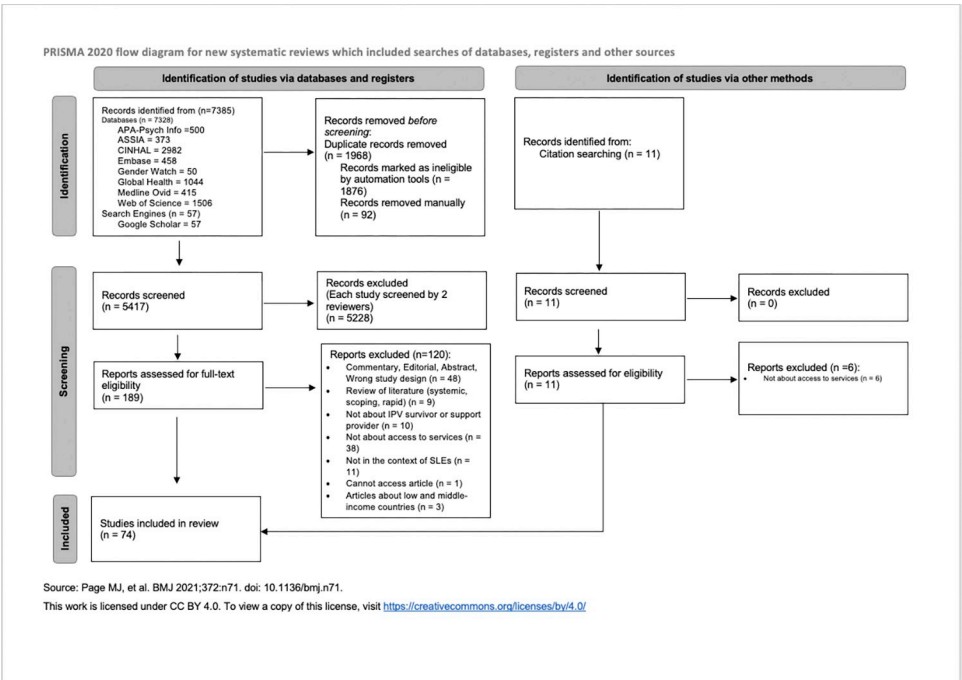

**Fig 1. PRISMA flow chart.**

(n = 42), originated from North America, specifically Canada and the United States. This is followed by 30% (n = 22) of the studies from Europe, and the remainder, 13% (n = 10), from Australia, New Zealand, and Japan. Regarding the perspectives of the participants, the majority, 62% (n = 46), involved formal service providers, 23% (n = 17) focused solely on survivors of IPV, and 14% (n = 10) examined both survivors and formal service providers. Only a marginal 1% (n = 1) of the studies discussed informal supports such as family, friends, or bystanders. In terms of the context of stressful life events (SLEs), 92% (n = 68) of the articles addressed experiences during the COVID-19 pandemic, while the remaining 8% (n = 6) were distributed between studies on IPV during economic recessions (n = 2) and natural disasters such as floods, wildfires, hurricanes, snowstorms and earthquakes (n = 4). The methodological approaches varied: nearly 69% (n = 51) of the studies employed qualitative methods, 17% (n = 13) utilized multi-methods (qualitative and quantitative) and mixed methods, over 11% (n = 8) were strictly quantitative, and two studies, making up 3% (n = 2) of the total, were case studies.

## Barriers and facilitators to accessing IPV supports during stressful life events (SLEs)

Nine themes emerged through thematic analysis while exploring what is known in the literature about access to services by individuals who experience IPV during SLEs in high-income countries from the perspectives of IPV Survivors as well as formal and informal IPV support providers. Key themes include (i) impacts of SLEs on health and well-being of IPV Survivors, (ii) considerations in the violence against women (VAW) sector during SLEs, (iii) capacity to provide and access support services following SLEs, (iv) service providers workload and stress during SLEs, (v) issues of isolation, power dynamics and control during SLEs for IPV Survivors, (vi) navigating the transition to virtual service amidst the pandemic, (vii) privacy, safety

**Table 1. Summary characteristics of included articles.**

| Study ID | Geography (country, region etc.) | Title | Aim of study | Type of Study | Type of Stressful Life Event | Participant Type | Type of supports being accessed or provided | Total Sample |
|---|---|---|---|---|---|---|---|---|
| | | | | NORTH AMERICA | | | | |
| **Bomsta & Kerr (2023) [37]** | United States | Organizational Resilience of Intimate Partner Abuse Nonprofits During the COVID-19 Pandemic | To examine eight IPA nonprofits in a Midwestern state during the COVID-19 pandemic, interviewing both managers and frontline staff. | Qualitative research | COVID-19 pandemic | Service/ support providers | Formal | 18 |
| **Brown et al. (2023) [38]** | United States | Shelter in the Storm: A Battered Women's Shelter and Catastrophe | To investigate the impact of Hurricane Katrina on a New Orleans shelter, extract lessons from the experience to guide shelters facing future disasters and explore how an agency with specific policies and structures navigated and recovered from the overwhelming effects of Hurricane Katrina. | Qualitative research | Natural disaster (hurricane, flood, earthquake) | Service/ support providers | Formal | 20 |
| **Burd et al. (2023) [39]** | Canada | "Our services are not the same": the impact of the COVID-19 pandemic on care interactions in women's shelters | To provide further understanding on how COVID-19 pandemic protocols impacted care interactions within women's shelters in Ontario, Canada. | Qualitative research | COVID-19 pandemic | Both IPV and support providers | Formal | 50 |
| **Elliott et al. (2022) [40]** | United States | Survivors' Concerns During the COVID-19 Pandemic: Qualitative Insights from the National Sexual Assault Online Hotline | To understand the pandemic-related experiences and concerns of sexual violence survivors by analyzing open-ended data collected via staff who interacted with users of the National Sexual Assault Online Hotline. | Qualitative research | COVID-19 pandemic | Survivors of IPV | Formal | 470 |
| **Enarson (1999) [41]** | United States and Canada | Violence Against Women in Disasters: A Study of Domestic Violence Programs in the United States and Canada | To investigate organizational readiness, impact and response of domestic violence programs in Canada and US. | Qualitative and Quantitative research | Natural disasters (hurricane, flood, earthquake) | Service/ support providers | Formal | 77 |
| **Engleton et al. (2022) [42]** | United States | Sexual assault survivors' engagement with advocacy services during the COVID-19 pandemic | To understand how the COVID-19 pandemic transformed survivors' engagement with sexual assault advocacy services. | Qualitative research | COVID-19 pandemic | Service/ support providers | Formal | 12 |
| **Fleury-Steiner et al. (2022) [43]** | United States | Online Guidance for Domestic Violence Survivors and Service Providers: A COVID-19 Content Analysis | To examine the prevalence and type of information U.S. national and state/territorial domestic violence (DV) organizations provided to direct service providers, to survivors, and to racially and culturally specific communities through their websites. | Qualitative research | COVID-19 pandemic | Service/ support providers | Formal | 67 |

*(Continued)*

**Table 1.** (Continued)

| Study ID | Geography (country, region etc.) | Title | Aim of study | Type of Study | Type of Stressful Life Event | Participant Type | Type of supports being accessed or provided | Total Sample |
|---|---|---|---|---|---|---|---|---|
| Fothergill (1999) [44] | United States | An Exploratory Study of Woman Battering in the Grand Forks Flood Disaster: Implications for Community Responses and Policies | To explore what intimate partner violence means to women in the face of natural disasters using the 1997 floods in Grand Forks as a case study. | Case study | Natural disaster (hurricane, flood, earthquake) | Service/ support providers | Formal | 2 |
| Fuchsel (2024) [45] | United States | Immigrant Latinas' Experiences with Intimate Partner Violence, Access to Services, and Support Systems During a Global Health Crisis (COVID-19) | To examine immigrant Latina's help-seeking behaviors, types of support systems, and access to intimate partner violence (IPV) services during a global health crisis (COVID-19) at a community-based agency. | Qualitative research | COVID-19 pandemic | Survivors of IPV | Formal and Informal | 19 |
| Garcia et al. (2022) [46] | United States | The Impact of the COVID-19 Pandemic on Intimate Partner Violence Advocates and Agencies | To explore (1) personal challenges and resilience working as IPV advocates during the COVID-19 pandemic; (2) how agencies adapted to the pandemic to support IPV survivors and advocates; and (3) specific needs and challenges of culturally-specific agencies. | Qualitative research | COVID-19 pandemic | Service/ support providers | Formal | 53 |
| Ghidei et al. (2022) [47] | Canada | Perspectives on delivering safe and equitable trauma-focused intimate partner violence interventions via virtual means: A qualitative study during COVID-19 pandemic | To qualitatively describe the challenges experienced by service providers with virtually delivering IPV services that are safe, equitable, and accessible for their diverse clients during the COVID-19 pandemic. | Qualitative research | COVID-19 pandemic | Service/ support providers | Formal | 24 |
| Haag et al. (2022) [48] | Canada | The Shadow Pandemic: A Qualitative Exploration of the Impacts of COVID-19 on Service Providers and Women Survivors of Intimate Partner Violence and Brain Injury | To explore the impact of the COVID-19 pandemic on survivors and service providers. | Qualitative research | COVID-19 pandemic | Both IPV and support providers | Formal | 18 |
| Hendrix (2021) [49] | United States | "Boiling water but there's no pop-off valve" Health care professionals' perceptions of the effects of COVID-19 on Intimate Partner Violence | To understand the impacts of COVID-19, including the impacts of movement-related restrictions such as shelter-in-place, quarantine, and isolation orders, on experiences of IPV from the perspective of health care professionals. | Qualitative research | COVID-19 pandemic | Service/ support providers | Formal | 8 |
| Krishnamurti et al. (2021) [50] | United States | Mobile Remote Monitoring of Intimate Partner Violence Among Pregnant Patients During the COVID-19 Shelter-In-Place Order: Quality Improvement Pilot Study | To examine cases of IPV that were reported on a prenatal care app before and during the implementation of COVID-19 shelter-in-place mandates. | Quantitative | COVID-19 pandemic | Survivors of IPV | Formal | 415 |

(*Continued*)

**Table 1.** (Continued)

| Study ID | Geography (country, region etc.) | Title | Aim of study | Type of Study | Type of Stressful Life Event | Participant Type | Type of supports being accessed or provided | Total Sample |
|---|---|---|---|---|---|---|---|---|
| **Lapierre et al. (2022) [51]** | Canada | 'We have tried to remain warm despite the rules.' Domestic violence and COVID-19: implications for shelters' policies and practices | To investigate the impacts of the COVID-19 pandemic on domestic violence shelters' policies and practices. | Qualitative research | COVID-19 pandemic | Service/ support providers | Formal | 9 |
| **Leat et al. (2024) [52]** | United States | Living in an Intimate Partner Violence Shelter During a Pandemic: Perspectives from Advocates and Survivors | To fully understand the impact of adaptations during the coronavirus pandemic, the objective is to interview shelter staff and residents to capture their experiences of living and working together during the summer of 2020, thereby assessing how changes in policy and procedures within shelters have affected both survivors and advocates. | Qualitative research | COVID-19 pandemic | Both IPV and support providers | Formal | 10 |
| **Leigh et al. (2022) [53]** | United States | "Are you safe to talk?": Perspectives of Service Providers on Experiences of Domestic Violence During the COVID-19 Pandemic | To understand the factors shaping reported trends in domestic violence during the pandemic. | Qualitative research | COVID-19 pandemic | Service/ support providers | Formal | 32 |
| **Luebke et al. (2023) [54]** | United States | Barriers Faced by American Indian (AI) Women in Urban Wisconsin in Seeking Help Following an Experience of Intimate Partner Violence | To better understand the context of IPV and help-seeking behaviors for urban AI women after experiences with IPV. | Qualitative research | COVID-19 pandemic | Survivors of IPV | Formal and Informal | 34 |
| **MacGregor et al. (2022) [55]** | Canada | Experiences of Women Accessing Violence Against Women Outreach Services in Canada During the COVID-19 Pandemic: A Brief Report | To study women's experiences with VAW services in the first stages of the pandemic and describe their fears and concerns. | Quantitative | COVID-19 pandemic | Survivors of IPV | Formal and Informal | 49 |
| **Mantler et al. (2022) [56]** | Canada | Navigating multiple pandemics: A critical analysis of the impact of COVID-19 policy responses on gender-based violence services | To explore the effects of the COVID-19 pandemic, and policy responses to it, on the provision of gender-based violence (GBV) services, as described by staff and executive directors in GBV organizations in Ontario, Canada | Qualitative research | COVID-19 pandemic | Service/ support providers | Formal | 50 |
| **Mantler et al. (2024) [57]** | Canada | Resilience is more than Nature: An Exploration of the Conditions that Nurture Resilience Among Rural Women who have Experienced IPV | To explore how rural women experiencing IPV cultivate resilience, the ability to survive and thrive in spite of adversity, and the environmental of IPV. | Qualitative research | COVID-19 pandemic | Both IPV and support providers | Formal | 12 |

(*Continued*)

**Table 1.** (Continued)

| Study ID | Geography (country, region etc.) | Title | Aim of study | Type of Study | Type of Stressful Life Event | Participant Type | Type of supports being accessed or provided | Total Sample |
|---|---|---|---|---|---|---|---|---|
| Michaelsen et al. (2022) [58] | Canada | Service provider perspectives on how COVID-19 and pandemic restrictions have affected intimate partner and sexual violence survivors in Canada: a qualitative study | To explore the perspectives of intimate partner and sexual violence workers across Canada on how the COVID-19 pandemic has affected the survivors with whom they work. | Qualitative research | COVID-19 pandemic | Service/ support providers | Formal | 17 |
| Montesanti et al. (2022) [59] | Canada | Examining organization and provider challenges with the adoption of virtual domestic violence and sexual assault interventions in Alberta, Canada, during the COVID-19 pandemic | To understand the challenges in the implementation of virtual and remote-based services and interventions from the perspective of organizational leaders and service providers from the antiviolence sector in Alberta during the COVID-19 pandemic | Qualitative research | COVID-19 pandemic | Service/ support providers | Formal | 24 |
| Nahar et al. (2023) [60] | United States | Challenges and Adaptations Experienced by Intimate Partner Violence Service Providers During COVID-19 Pandemic | To explore the experiences of IPV service providers during this pandemic | Mixed Methods | COVID-19 pandemic | Service/ support providers | Formal | 15 |
| Nnawulezi and Hacskaylo (2022) [61] | United States | Identifying and Responding to the Complex Needs of Domestic Violence Housing Practitioners at the Onset of the COVID-19 Pandemic. | To systematically document the immediate impact of COVID-19 on program operations and service provision within the domestic violence field, address the challenges in negotiating the safety paradox and responding to survivors' immediate housing needs, and provide practical recommendations for managing both the short and long-term effects of the pandemic. | Qualitative and Quantitative research | COVID-19 pandemic | Service/ support providers | Formal | 840 |
| Oswald et al. (2023) [62] | United States | American Women's Experiences with Intimate Partner Violence during the Start of the COVID-19 Pandemic: Risk Factors and Mental Health Implications | To investigate American women's experiences with intimate partner violence (IPV) during the early months of the COVID-19 pandemic, aiming to understand how specific COVID-19 related stress factors are associated with IPV experiences while considering vulnerability risk factors such as socioeconomic status, community type, and sexual orientation. | Quantitative | COVID-19 pandemic | Survivors of IPV | Formal | 1168 |

(*Continued*)

**Table 1.** (Continued)

| Study ID | Geography (country, region etc.) | Title | Aim of study | Type of Study | Type of Stressful Life Event | Participant Type | Type of supports being accessed or provided | Total Sample |
|---|---|---|---|---|---|---|---|---|
| **Ragavan et al. (2022) [63]** | United States | The Impact of the COVID-19 Pandemic on the Needs and Lived Experiences of Intimate Partner Violence Survivors in the United States: Advocate Perspectives | To examine the impact of the COVID-19 pandemic on family violence and related service provision, this study conducted qualitative interviews with U.S.-based IPV advocates. The aim is to explore their perspectives on the stressors faced by IPV survivors during the pandemic, the strategies survivors employed to keep themselves safe, and the unique challenges and protective factors for IPV survivors from marginalized communities. | Qualitative research | COVID-19 pandemic | Service/ support providers | Formal | 53 |
| **Rahman et al. (2022) [64]** | United States | Intimate Partner Violence and the COVID-19 Pandemic | To describe the effects of the COVID-19 pandemic and associated practice shifts on consultation and referral patterns of an intimate partner violence program at a large, urban children's hospital. | Quantitative | COVID-19 pandemic | Survivors of IPV | Formal | 295 |
| **Randell et al. (2023) [65]** | United States | COVID-19 Pandemic Impact on United States Intimate Partner Violence Organizations: Administrator Perspectives | To explore administrative perspectives on the impacts of the COVID-19 pandemic on United States regional and national IPV service organizations | Qualitative research | COVID-19 pandemic | Service/ support providers | Formal | 35 |
| **Ravi et al. (2022) [66]** | United States | Survivors' experiences of intimate partner violence and shelter utilization during COVID-19 | To explore the impact of COVID-19 on female survivors' experiences of violence and IPV services. | Qualitative research | COVID-19 pandemic | Survivors of IPV | Formal | 10 |
| **Safar et al. (2023) [67]** | Canada | Exploring Coping Strategies Among Older Women Who Have Experienced Intimate Partner Violence During COVID-19 | To investigate coping among older women in Ontario experiencing intimate partner violence (IPV) during COVID-19. | Qualitative research | COVID-19 pandemic | Survivors of IPV | Formal and Informal | 12 |
| **Sapire et al. (2022) [68]** | United States | COVID-19 and gender-based violence service provision in the United States | To examine how the existing GBV funding and policy landscape, COVID-19, and resulting state policies in the first six-months of the pandemic affect GBV health service provision in the U.S. | Mixed Methods | COVID-19 pandemic | Service/ support providers | Formal | 77 |
| **Segura et al. (2022) [69]** | United States | Rethinking Dating and SEXUAL Violence Prevention for Youth During the Pandemic: Examining Program Feasibility and Acceptability | To understand how youth perceive learning Sexual and dating violence (SDV) prevention in an online environment. | Mixed Methods | COVID-19 pandemic | Survivors of IPV | Formal | 12 |

(*Continued*)

**Table 1.** (Continued)

| Study ID | Geography (country, region etc.) | Title | Aim of study | Type of Study | Type of Stressful Life Event | Participant Type | Type of supports being accessed or provided | Total Sample |
|---|---|---|---|---|---|---|---|---|
| **Shyrokonis et al. (2024)** [70] | United States | Help-Seeking and Service Utilization Among Survivors of Intimate Partner Violence in Michigan During the COVID-19 Pandemic | To explore formal and informal intimate partner violence (IPV) service use among women and transgender/nonbinary individuals in the state of Michigan during the COVID-19 pandemic | Quantitative | COVID-19 pandemic | Survivors of IPV | Formal and Informal | 179 |
| **Storer and Nyerges (2023)** [71] | United States | The Rapid Uptake of Digital Technologies at Domestic Violence and Sexual Assault Organizations During the COVID-19 Pandemic | To explore domestic violence and sexual assault (DV/SA) service providers' perceptions of how their organizations responded to the pandemic. | Qualitative research | COVID-19 pandemic | Service/ support providers | Formal | 20 |
| **Toccalino et al. (2022)** [72] | Canada | Addressing the Shadow Pandemic: COVID-19 Related Impacts, Barriers, Needs, and Priorities to Health Care and Support for Women Survivors of Intimate Partner Violence and Brain Injury | To assess and address the care needs of women survivors of IPV who present with traumatic brain injury (TBI) to identify key needs, facilitators, and barriers to care specific to the COVID-19 pandemic and more broadly, and to generate resources and principles for identification, clinical care, and support for healthcare practitioners treating women exposed to IPV-related TBI. | Qualitative | COVID-19 pandemic | Both IPV and support providers | Formal | 27 |
| **VothSchrag et al. (2023)** [73] | United States | So many extra safety layers: Virtual service provision and implementing social distancing in interpersonal violence service agencies during COVID-19 | To investigate the strategies community-based interpersonal violence services employed to comply with social distancing guidelines during the COVID-19 pandemic, and to understand the experiences of anti-violence service professionals tasked with implementing these adaptations. | Mixed methods | COVID-19 pandemic | Service/ support providers | Formal | 33 |
| **Williams et al. (2021)** [74] | United States | Provider perspectives on the provision of safe, equitable, trauma-informed care for intimate partner violence survivors during the COVID-19 pandemic: a qualitative study | To understand and characterize the impact of the pandemic on delivery of IPV care in Boston. | Qualitative research | COVID-19 pandemic | Service/ support providers | Formal | 18 |
| **Wood et al. (2022)** [75] | United States | "Don't Know where to Go for Help": Safety and Economic Needs among Violence Survivors during the COVID-19 Pandemic | To understand the health, safety, and economic impacts of the COVID-19 pandemic on people that are experiencing or have previously experienced violence, stalking, threats, and/or abuse. | Mixed Methods | COVID-19 pandemic | Survivors of IPV | Formal and Informal | 53 |

(*Continued*)

**Table 1.** (Continued)

| Study ID | Geography (country, region etc.) | Title | Aim of study | Type of Study | Type of Stressful Life Event | Participant Type | Type of supports being accessed or provided | Total Sample |
|---|---|---|---|---|---|---|---|---|
| **Wright et al. (2022) [76]** | United States | The Impact of COVID-19 Restrictions on Victim Advocacy Agency Utilization Across Pennsylvania | To examine the impact of COVID-19 restrictions on the utilization of Victim Assistance Agencies (VAAs) in Pennsylvania, using 2019–2020 data, in order to develop solutions that maintain access to services despite limitations on face-to-face interactions. | Qualitative and Quantitative research | COVID-19 pandemic | Service/ support providers | Formal | 49 |
| **Wyckoff et al. (2023) [77]** | United States | "COVID gave him an opportunity to tighten the reins around my throat": perceptions of COVID-19 movement restrictions among survivors of intimate partner violence | To understand the impacts of COVID-19, including the impacts of movement restrictions (i.e., shelter in place orders, quarantine, isolation orders) on experiences of IPV from the perspective of survivors. | Qualitative research | COVID-19 pandemic | Survivors of IPV | Formal | 10 |
| **Yakubovich et al. (2023) [78]** | Canada | Recommendations for Canada's National Action Plan to end gender-based violence: perspectives from leaders, service providers and survivors in Canada's largest city during the COVID-19 pandemic | To analyze the perspectives of violence-against-women leaders, service providers and survivors in the Greater Toronto Area during the COVID-19 pandemic to generate recommendations for the federal government's proposed National Action Plan to End Gender-Based Violence. | Qualitative research | COVID-19 pandemic | Both IPV and support providers | Formal | 18 |
| **EUROPE** | | | | | | | | |
| **Anitha and Gill (2022) [79]** | United Kingdom | Domestic violence during the pandemic: 'By and for' frontline practitioners' mediation of practice and policies to support racially minoritised women | To explore the responses of frontline female practitioners from domestic violence and abuse (DVA) services for racially minoritized women in England and Wales to the challenges posed by the COVID-19 crisis, with a focus on their shared community background with the women they support. | Qualitative research | COVID-19 pandemic | Service/ support providers | Formal | 26 |
| **Briones-Vozmediano et al. (2014) [80]** | Spain | Economic crisis, immigrant women and changing availability of intimate partner violence services: a qualitative study of professionals' perceptions in Spain | To explore intimate partner violence (IPV) service providers' perceptions of the impact of the current economic crisis on these resources in Spain and on their capacity to respond to immigrant women's needs experiencing IPV. | Qualitative research | Economic recession | Service/ support providers | Formal | 39 |

(*Continued*)

**Table 1.** (Continued)

| Study ID | Geography (country, region etc.) | Title | Aim of study | Type of Study | Type of Stressful Life Event | Participant Type | Type of supports being accessed or provided | Total Sample |
|---|---|---|---|---|---|---|---|---|
| Caridade et al. (2021) [81] | Portugal | Remote support to victims of violence against women and domestic violence during the COVID-19 pandemic | To characterize the type of support provided to victims of violence against women and domestic violence (VAWDV) during the first lockdown and assess the training of professionals to use remote support (RS). | Quantitative | COVID-19 pandemic | Service/ support providers | Formal | 196 |
| Castellanos-Torres et al. (2023) [82] | Spain | COVID-19 and sexual violence against women: a qualitative study about young people and professionals' perspectives in Spain | To investigate sexual violence (SV) during the COVID-19 lockdown among young people in Spain and to examine SV-related services from the perspectives of professionals and youth involved in addressing SV and other forms of violence against women. | Qualitative research | COVID-19 pandemic | Both IPV and support providers | Formal | 15 |
| Cunha et al. (2024) [83] | Portugal | Domestic violence professionals in Portuguese shelters: navigating challenges amidst the COVID-19 pandemic | To investigate the experiences of professionals working in shelters in Portugal and their responses to domestic violence within the context of the COVID-19 pandemic. | Qualitative research | COVID-19 pandemic | Service/ support providers | Formal | 30 |
| Desai et al. (2022) [84] | United Kingdom | The Experiences of Post-Separation Survivors of Domestic Violence During the Covid-19 Pandemic: Findings from a Qualitative Study in the United Kingdom | To investigate the experiences of separated DV survivors (all women) during the Covid-19 pandemic. | Qualitative research | COVID-19 pandemic | Survivors of IPV | Formal | 21 |
| Elvey et al. (2022) [85] | United Kingdom | A hospital-based independent domestic violence advisor service: demand and response during the Covid-19 pandemic | To explore and understand a hospital-based independent domestic violence advisor service activity and response during the pandemic. | Mixed Methods | COVID-19 pandemic | Service/ support providers | Formal | 11 |
| Emsley et al. (2023) [86] | United Kingdom | General practice as a place to receive help for domestic abuse during the COVID-19 pandemic: a qualitative interview study in England and Wales | To understand the patient perspective of seeking and receiving help for domestic violence and abuse (DVA) in general practice during the COVID-19 pandemic. | Qualitative research | COVID-19 pandemic | Both IPV and support providers | Formal | 13 |
| Gill and Anitha (2023) [87] | United Kingdom | The nature of domestic violence experienced by Black and minoritised women and specialist service provision during the COVID-19 pandemic: practitioner perspectives in England and Wales | To understand the contours of what has been termed a "dual pandemic" in the UK: twin crises of increasing domestic violence and abuse (DVA) alongside the spread of COVID-19, both of which have disproportionately affected Black and minoritised communities. | Qualitative research | COVID-19 pandemic | Service/ support providers | Formal | 26 |

(*Continued*)

**Table 1.** (Continued)

| Study ID | Geography (country, region etc.) | Title | Aim of study | Type of Study | Type of Stressful Life Event | Participant Type | Type of supports being accessed or provided | Total Sample |
|---|---|---|---|---|---|---|---|---|
| Golenko and Ritossa (2022) [88] | Republic of Croatia | The role of civil society in a time of pandemic: bridging the gap between official policies and information needs of victims of family violence | To investigate the problems encountered by employees of NGOs in accessing and using information from the immediate environment during COVID-19, especially relevant information for preserving and protecting the rights of persons exposed to family violence and potentially other vulnerable groups. | Qualitative research | COVID-19 pandemic | Service/ support providers | Formal | 21 |
| Gregory and Williamson (2021) [89] | United Kingdom | 'I think it just made everything very much more intense': A qualitative secondary analysis exploring the role of friends and family providing support to survivors of domestic abuse during the covid-19 pandemic | To highlight narratives from people providing informal support to domestic abuse (DA) survivors during the pandemic, with the following research questions: (i) How has the pandemic changed the ways that informal supporters are able to offer support, (ii) How has the pandemic changed situations of domestic abuse. | Qualitative research | COVID-19 pandemic | Family and Friends | Informal | 18 |
| Holt et al. (2023) [90] | Ireland | Social Workers Response to Domestic Violence and Abuse during the COVID-19 Pandemic | To establish the extent and nature of domestic violence and abuse in social work practice during the initial period of COVID-19 "lockdown" restrictions. | Qualitative and Quantitative research | COVID-19 pandemic | Service/ support providers | Formal | 120 |
| McKinlay et al. (2023) [91] | United Kingdom | How did UK social distancing restrictions affect the lives of women experiencing intimate partner violence during the COVID-19 pandemic? A qualitative exploration of survivor views | To learn more about how the lives of women who experience intimate partner violence were impacted by social distancing restrictions. | Qualitative research | COVID-19 pandemic | Survivors of IPV | Formal and Informal | 18 |
| Otero-Garcia et al. (2018) [92] | Spain | A qualitative study on primary health care responses to intimate partner violence during the economic crisis in Spain | To explore how health professionals perceived the effect of the economic crisis and associated austerity measures on the detection of and cares for intimate partner violence(IPV) in primary care in Spain. | Qualitative research | Economic recession | Service/ support providers | Formal | 145 |
| Pedersen et al. (2023) [93] | Scotland | Partnerships between police and GBV service providers in remote, rural, and island communities in northern Scotland before and during COVID-19 | To understand the reality of partnership working between police and third-sector organizations. | Qualitative research | COVID-19 pandemic | Service/ support providers | Formal | 15 |

**Table 1.** (Continued)

| Study ID | Geography (country, region etc.) | Title | Aim of study | Type of Study | Type of Stressful Life Event | Participant Type | Type of supports being accessed or provided | Total Sample |
|---|---|---|---|---|---|---|---|---|
| Pedersen et al. (2023) [94] | Scotland | Supporting victims of domestic violence in rural and island communities during COVID-19: the impact of the pandemic on service providers in Northeast Scotland and Orkney | To investigate the impact of the pandemic on the work of organisations supporting women who are experiencing gender-based violence in three distinct locations in Scotland. | Qualitative research | COVID-19 pandemic | Service/ support providers | Formal | 12 |
| Petersson and Hansson (2022) [95] | Sweden | Social Work Responses to Domestic Violence During the COVID-19 Pandemic: Experiences and Perspectives of Professionals at Women's Shelters in Sweden. | To explore how professionals at women's shelters experienced, understood, and responded to their clients' help-seeking during a period in which they were sharing a collective trauma with their clients. | Qualitative research | COVID-19 pandemic | Service/ support providers | Formal | 14 |
| Puigvert et al. (2021) [96] | Spain | BraveNet Upstander Social Network against Second Order of Sexual Harassment | To demonstrate the relevance of bringing Second Order of Sexual Harassment (SOSH) into the domestic violence spectrum that occurred during COVID-19 confinement. | Quantitative | COVID-19 pandemic | Service/ support providers | Formal | 23 |
| Speed et al. (2020) [97] | United Kingdom | Stay Home, Stay Safe, Save Lives? An Analysis of the Impact of COVID-19 on the Ability of Victims of Gender-based Violence to Access Justice | To explore the impact of COVID-19 on the ability of GBV services to operate and support victims in the current climate. | Qualitative research | COVID-19 pandemic | Service/ support providers | Formal | 52 |
| Stanley et al. (2022) [98] | United Kingdom | Developing new portals to safety for domestic abuse survivors in the context of the pandemic | To examine the emergence and implementation of community touch-points established in the UK during the COVID-19 pandemic for victims/survivors of domestic abuse (DA). | Case Study | COVID-19 pandemic | Both IPV and support providers | Formal and Informal | ~60 |
| Thiara and Roy (2022) [99] | United Kingdom | 'The disparity is evident': COVID-19, violence against women and support for Black and minoritised survivors | To explore the ways in which survivors experienced and responded to expanded forms of abuse, the ways in which organisations reshaped their provision, and the responses they and women received from mainstream service providers and the greater levels of intersectional advocacy that were required. | Qualitative research | COVID-19 pandemic | Service/ support providers | Formal | 14 |

(*Continued*)

**Table 1.** (Continued)

| Study ID | Geography (country, region etc.) | Title | Aim of study | Type of Study | Type of Stressful Life Event | Participant Type | Type of supports being accessed or provided | Total Sample |
|---|---|---|---|---|---|---|---|---|
| **Vives-Cases et al. (2021) [100]** | Spain | Coping with intimate partner violence and the COVID-19 lockdown: The perspectives of service professionals in Spain | To analyze the impact of COVID-19 confinement on IPV, their related services, and working conditions from the perspective of the professionals of different sectors, including social services, health services, the police, and legal support, both on behalf of public institutions as well as the third sector in Spain. | Qualitative research | COVID-19 pandemic | Service/ support providers | Formal | 47 |
| **AUSTRALIA, NEW ZEALAND & JAPAN** | | | | | | | | |
| **Alston et al. (2022) [101]** | Australia and New Zealand | Australian and New Zealand social workers adjusting to the COVID-19 pandemic | To explore the impact of COVID-19 on social work practice in Australia and New Zealand, focusing on how social workers have managed to safeguard vulnerable community members during lockdown, the main issues presented by clients, and the effects of the pandemic on social workers themselves. | Qualitative and Quantitative research | COVID-19 pandemic | Service/ support providers | Formal | 208 |
| **Clarke et al. (2023) [102]** | Australia | Resilience and Resistance in the Community Sector: Organisational Challenges and Responses by the Australian DFV Sector in the Time of COVID-19 Pandemic | To examine the impacts of COVID-19 on the domestic and family violence (DFV) service sector and the adaptations and innovations that emerged in response. | Qualitative and Quantitative research | COVID-19 pandemic | Service/ support providers | Formal | 362 |
| **Coram et al. (2021) [103]** | Australia | Community service sector resilience and responsiveness during the COVID-19 pandemic: The Australian experience | To assess the community sector's immediate reaction to the pandemic. | Qualitative and Quantitative research | COVID-19 pandemic | Service/ support providers | Formal | 279 |
| **Cortis et al. (2021) [104]** | Australia | Adapting Service Delivery during COVID-19: Experiences of Domestic Violence Practitioners | To examine service responses in Australia, exploring practitioners' accounts of adapting service delivery models in the early months of the pandemic. | Qualitative research | COVID-19 pandemic | Service/ support providers | Formal | 100 |
| **Fogarty et al. (2022) [105]** | Australia | Providing therapeutic services to women and children who have experienced intimate partner violence during the COVID-19 pandemic: Challenges and learnings | To explore parents' experiences of participating in a parent-child tele health intervention during the COVID-19 pandemic. The study also explored clinicians' experiences of delivering the service, including key strengths and challenges. | Qualitative research | COVID-19 pandemic | Both IPV and support providers | Formal | 14 |

(*Continued*)

**Table 1.** (Continued)

| Study ID | Geography (country, region etc.) | Title | Aim of study | Type of Study | Type of Stressful Life Event | Participant Type | Type of supports being accessed or provided | Total Sample |
|---|---|---|---|---|---|---|---|---|
| **Houghton et al., (2010) [106]** | New Zealand | "If There Was a Dire Emergency, We Never Would Have Been Able to Get in There": Domestic Violence Reporting and Disasters | This work examines domestic violence reporting in a community in New Zealand struck by a snowstorm. | Qualitative research | Natural disaster (hurricane, flood, earthquake) | Survivors of IPV | Formal | 7 |
| **Katou and Kataoka (2022) [107]** | Japan | Intimate partner violence and the situation of women experiencing intimate partner violence during the COVID-19 pandemic: A qualitative study of Japanese clinician views | To clarify IPV and the situation of women experiencing IPV during theCOVID-19 pandemic in Japan. | Qualitative research | COVID-19 pandemic | Service/ support providers | Formal | 5 |
| **Pfitzner et al. (2022) [108]** | Australia | When staying home isn't safe: Australian practitioner experiences of responding to intimate partner violence during COVID-19 restrictions | To explore the impact of remote service delivery on practitioner mental health and wellbeing and the quality of care provided. | Qualitative and Quantitative | COVID-19 pandemic | Service/ support providers | Formal | 166 |
| **Pfitzner et al. (2022) [109]** | Australia | Responding to women experiencing domestic and family violence during the COVID-19 pandemic: Exploring experiences and impacts of remote service delivery in Australia | To explore the professional experiences of practitioners supporting women experiencing violence during the pandemic | Qualitative research | COVID-19 pandemic | Service/ support providers | Formal | 117 |
| **Pfitzner and McGowen (2023) [110]** | Australia | Locked out or let in? Learning from victim-survivors' remote help-seeking experiences during COVID-19 | To determine the experiences and expertise of victim-survivors, using findings from an Australian study that investigated victim-survivors' use of DFV services during lockdowns. | Qualitative research | COVID-19 pandemic | Survivors of IPV | Formal | |

and confidentiality, (vii) intersectionality and issues faced by racialized and marginalized communities, and (ix) financial strain and abuse.

**Theme 1: Impacts of SLEs on health and well-being of survivors of IPV and the increased risk of IPV.** During SLEs, Survivors of IPV often faced a notable decline in their health and well-being, influenced by various factors, such as increased severity and frequency of IPV, social isolation, and limited access to formal and informal support networks While most literature in this review focuses on the COVID-19 pandemic, Enarson (1999) previously showed how disasters, like floods, can diminish Survivors' emotional and physical health due to isolation and lack of support [41]. This study further emphasized how environmental disasters place considerable strain on relationships by compounding the difficulties of replacing lost possessions, securing housing, and finding employment, all of which heighten the risk of intimate partner violence (IPV). Three other studies focused on recessions and environmental disasters, predominantly from the perspective of service providers [38, 80, 92].

Survivors encountered increased mental and physiological stressors throughout the pandemic, particularly during lockdown periods. As described in multiple studies, lockdown measures precipitated lifestyle changes (i.e., increased caregiving responsibilities, cooking,

cleaning, virtual schooling, workplace transitions, employment loss/change) marked by physical and social isolation [40, 43, 45, 61, 67, 71, 72, 74, 77–79, 84, 87, 88, 100, 102, 107, 110].

Many Survivors found themselves isolated at home with their perpetrator, lacking formal or informal support for themselves and their children. These changes, in turn, contributed to a decline in mental health, heightened anxiety, increased economic stress, increased alcohol consumption by perpetrators, increased instances and severity of abuse, and diminished motivation to seek help, exacerbating pre-existing trauma [67, 72, 75, 99]. Consequently, coping mechanisms such as alcohol or illicit substance use, engaging in sex work, or remaining in abusive relationships due to loneliness became more prevalent during the COVID-19 pandemic [77, 91].

Leigh (2022) revealed how limited access to services and reduced help-seeking behaviors stemmed from the need to juggle competing survival priorities such as job loss, childcare responsibilities, and caring for loved ones affected by COVID-19 [53]. Several studies reported observing an increase in violence, or the risk of violence and decreased mental health [45, 49, 60, 77, 79, 82, 83, 86, 99]. Elliott (2022) noted the heightened risk of violence, stating even Survivors not currently experiencing abuse feared pandemic-related stay-at-home mandates and associated stressors (e.g., COVID-19 infection, job loss) would precipitate new abuse [40]. Moreover, researchers in the United Kingdom observed the dynamics of how merging families to care for children and elderly relatives heightened stress and the risk of multi-perpetrator domestic violence and abuse (i.e., additional abuse from a perpetrator's family members), with racialized women being disproportionately affected [79, 87].

Various studies have detailed how survivors exhibited a decline in psychological well-being, including PTSD and depression, as well as symptoms such as flashbacks, nightmares, disturbed sleep patterns, anxiety, reduced self-esteem, neglect of personal care, and suicidal thoughts [84, 87, 99]. Haag (2022) emphasized the overlooked health concerns when virtual diagnosis and access to healthcare were not feasible during lockdowns, leading some to forego seeking health services [48]. Safar et al. (2023) and McKinlay et al. (2023) described how Survivors used physical or action-based coping strategies to manage stress and declining mental health, including cooking, baking, strategically avoiding the partner, taking on additional work, exercising, meditating and engaging in outdoor walks [67, 91]. From a service provider perspective, several authors illustrated how staff supported the well-being of clients by distributing care packages containing personal care and relaxation items during stressful periods, organizing distanced walking groups, and having 24-hour phone support to maintain client well-being contact during lockdowns [39, 99].

Mantler et al. (2024) discussed the inner strength and resilience of IPV Survivors during the pandemic, while Fothergill (1999) noted the personal growth and resilience that results from an SLE, in this case a flood [44]. They emphasized the importance of maintaining a positive mindset and resisting self-doubt to foster resilience, especially when supported by a close-knit community. Support helped mitigate mental health decline and isolation. The use of affirmations taught as a practical tool in counseling, reinforced survivors' self-worth and value [56].

**Theme 2: Considerations in the VAW sector during SLEs.** *The VAW sector–A neglected priority in emergency preparedness and SLEs.* The VAW sector has long grappled with challenges in delivering adequate support and resources to Survivors, even preceding the onset of events such as pandemics, economic downturns, or environmental catastrophes. Houghton et al. (2010) note that the VAW sector often faces a shortage of resources and limited capacity to adapt to climate crises [106]. Brown (2010) explained this struggle within the context of Hurricane Katrina in New Orleans, while Williams (2021) and Michaelsen (2022) expanded on how the already fragile infrastructure was exacerbated by the pandemic [38, 58, 74]. Pfitzner and McGowen (2023) noted that during the lockdowns, domestic and family violence

services were not classified as 'essential' services, which prevented them from remaining open for face-to-face consultations [110]. Stanley et al. (2022) investigated the development and use of community touch-points in the UK, such as pharmacies and banks, as potential frontline response sites for IPV survivors during COVID-19. They found that while these venues could offer necessary support, the significant training and resources needed by an already resource-strained sector present a substantial challenge [98].

Additionally, Enarson (1999) suggested an existing insensitivity towards domestic violence, which diminishes motivation and initiatives to include the sector in preparedness planning [41]. Governments across all levels have consistently neglected to acknowledge the pivotal role of VAW sectors in emergency preparedness planning, resulting in a dearth of recognition and support for these indispensable services during crises. This may be explained by the lack of recognition of shelters as priority facilities for housing and aiding a vulnerable population [41]. Briones-Vozmediano (2014) highlighted the relegation of IPV policy to a non-priority status during economic crises, subject to the whims of inconsistent political parties [80]. Consequently, economic downturns precipitate austerity measures, budget cuts, and diminished resources for IPV support, particularly for marginalized groups such as immigrant women. Otero-Garcia (2018) further illustrated how complementary resources to clinics, including shelters, social housing, criminal courts, and NGOs, are curtailed during periods of economic turmoil [92].

Articles focused on the pandemic also underscored the absence of established emergency preparedness plans or protocols to mount swift and effective responses [62, 74]. Findings from these articles recognized the importance of disaster readiness and emergency preparedness in addressing IPV during the pandemic, which encompassed factors such as regional subcultures of preparedness, prior disaster knowledge, governmental mandates, and personal networks [41, 49, 62, 110]. Furthermore, Enarson (1999) emphasized the imperative of survivor resilience and community integration into disaster planning to ensure the presence of structural support before crises occur [41]. Post-pandemic, Yakobovich et al. (2023) stressed the importance of the inclusion of VAW Survivor voices in policy planning [78].

*Precarious funding is an issue.* The chronic issue of unstable funding within the VAW sector has persistently hindered its capacity to deliver consistent and comprehensive support to Survivors of IPV. Multiple articles emphasize this challenge, particularly highlighting the vulnerability of the sector during SLEs. Specifically, in the aftermath of climate disasters such as hurricanes or earthquakes, physical infrastructures are often decimated, leading to the perception of services no longer existing which results in the cessation of funding [38]. Economic recessions exacerbate this predicament as austerity measures redirect funds away from social support programs, precipitating budget cuts and resource shortages [80, 92]. More recently, the onset of COVID-19 further compounded funding challenges, as the mechanisms and constraints surrounding service provider funding significantly affected stability, sustainability, and service provision. Insufficient and inconsistent funding impeded the transition to virtual services, and the short-term nature of COVID-19 funding failed to offer sustainable solutions [39, 59, 88, 103]. Consequently, the loss of funding precipitated staff layoffs, voluntary attrition, and a reduction in services [59, 61, 72]. Disparities emerged in accessing emergency funding during the pandemic, leading to competition among providers for limited resources, often with unsuccessful outcomes [68, 78]. Furthermore, inflexibility in funding allocation posed another challenge, as providers were mandated to allocate funds for specific purposes, such as personal protective equipment (PPE), at the expense of essential services, while some funds were inaccessible or difficult to obtain [42, 68]. Fundraising activities, typically relied upon to supplement core funding, were either suspended, delayed, or insufficient during emergencies [41, 59].

Several articles highlighted the importance of both governmental and private funding, as well as the need for flexibility in funding and service delivery to accommodate the increased severity and number of cases during SLEs. This included extending hours, hiring/training additional staff, enhancing advertising to raise awareness and provide education, transitioning to virtual services during the pandemic, providing hazard pay, additional mental health services, and adopting innovative programming [37, 61, 65, 78, 95]. Burd et al. (2023) further noted that community support through donations was vital in sustaining agency operations and facilitating the expansion of services [39].

*Coordination and collaboration are important.* The literature surrounding the pandemic highlighted the difficulty in or lack of coordination and collaboration within the VAW sector and across related systems (i.e., healthcare, social housing, legal services, and criminal justice), suggesting systemic hurdles to effectively address the diverse needs of IPV Survivors amidst crises.

The collegial partnerships among service providers dissipated as services transitioned online and service providers were busy adapting to the pandemic [104]. For example, Gregory (2021) discussed the challenges faced by service providers in coordinating support with friends and family to aid in safety planning in a virtual and locked-down environment [89]. At a broader sector level, Bomsta and Kerr (2023), Burd (2023), Garcia (2022), and Vives-Cases (2021) detailed the challenges service providers faced in collaborating with other agencies and support services during the pandemic. Limited interactions, mandated by public health measures and changes in service delivery, significantly impeded this fundamental aspect of their work. [37, 39, 46, 100]. Randell et al. (2023) addressed the challenges of maintaining connections with rural agencies that relied on travel, after travel had halted during the pandemic. This disruption harmed the partnerships essential for supporting survivors in rural areas [65]. Ghidei et al. (2022) addressed the dearth of coordination, collaboration, and transparency among VAW sector agencies during the pandemic, attributed to competition for funding [47]. In terms of coordination with other sectors, the COVID-19 pandemic disrupted the coordination along the housing continuum from emergency shelters to transitional housing and then social housing; there was increased demand for and reduced availability of housing, leading to difficulties in coordination in transitioning Survivors from emergency shelters [79]. Even if housing was available, Cunha et al. (2023) identified challenges in the coordination of processes for reintegration into society, such as setting up water service, acquiring furniture, or accessing social security [83]. Yakubovich et al. (2023) discussed the development of a continuum in housing support, ranging from emergency shelters to long-term supportive housing, and investment in transitional housing to provide stable environments for survivors [78]. In the context of climate crisis, Houghton et al., (2010) described the lack of communication and coordination between agencies to respond to IPV during the snow storm in New Zealand [106].

Facilitators included integration of IPV screening and support services with other essential services (general practice clinics, childcare services) to enhance continuity of care for consistent support and understanding of the Survivor experience [86, 108, 109]. Fuchsel (2024) emphasized the importance of supporting the overall well-being of Survivors through comprehensive wrap-around services, such as food, financial assistance, and counseling [45]. Similarly, Thiara and Roy (2022) stressed the necessity of enhancing wrap-around support for women with no recourse or access to public funds (i.e., not yet a citizen and have access to social welfare supports) including the provision of basic practical help with living essentials, to address their critical needs effectively [99]. Additionally, the necessity of interdisciplinary collaboration within the sector (i.e., community-based VAW organizations) and partnerships with other services (police, transportation, taxis, groceries, rideshares, landlords/housing, schools, healthcare professionals) was highlighted [37, 51, 60, 72, 78, 83, 86, 87, 90, 98, 100,

108]. Cunha et al. (2024) detailed how shelters collaborated with key organizations to ensure children had necessary resources for online schooling, while also securing essential supplies like PPE and food for shelters. This partnership provided comprehensive support, addressing both the educational needs of children and the operational requirements of the shelters [83].

*Inadequate communication and lack of public awareness.* Several studies examining the COVID-19 context highlighted issues of inadequate communication and insufficient public awareness, areas that were not specifically addressed in the research on other SLEs (environmental disasters and economic recessions). The absence of transparent communication regarding avenues for seeking assistance and ambiguity surrounding the availability of support services present obstacles for Survivors of IPV, compounding their already precarious situation during the pandemic [40]. Studies described the confusion stemming from inconsistent messaging regarding the closure of shelters and services despite many remaining operational during COVID-19; this was coupled with unclear guidelines, which impeded trust in available information [43, 51, 53, 54, 60, 65, 74, 75, 88]. In terms of public awareness, Mantler et al. (2024) highlighted service providers' concerns about systemic barriers stemming from a widespread lack of understanding of intimate partner violence (IPV) and the failure of systems like the police and criminal justice to adequately prioritize survivor safety [57]. Similarly, Yakubovich et al. (2023) addressed the insufficient awareness about IPV, including unclear definitions of what constitutes violence [78].

In the context of the COVID-19 pandemic, the continuously changing nature of public health regulations necessitated frequent revisions to institutional policies and procedures, rendering it challenging for Survivors to grasp the evolving situation and access necessary services [51, 74, 97], while also demanding daily adaptations from service providers to comply with these mandates [59]. McKinley et al. (2023) highlighted significant gaps in communication during the lockdowns concerning the ability to leave an abusive relationship; survivors were unclear about mandates and whether they could leave to access safe spaces [91]. In another example, Speed (2020) highlighted inadequate communication regarding case delays within legal systems and criminal trials, particularly concerning divorce and custody matters [97].

Scholars discussed the need to access clear information about options in the aftermath of SLEs, advocating for leveraging diverse media platforms to publicly disseminate information about support services [43, 49, 51, 53, 72, 86, 88, 97, 98, 100]. These initiatives include various media channels through local community and state-led campaigns (rather than general national level), collaborations with other entities, and innovative approaches to enhance public awareness. Authors underscored the criticality of providing Survivors of IPV with clear guidance on available options during lockdowns (or emergencies), advocating for media campaigns and establishing accessible helplines within safe environments [51, 97]. Stanley et al. (2022) suggested leveraging relationships with large public-facing companies, like Royal Mail in the U.K., to raise awareness about available support services for survivors IPV during emergencies [98].

Pedersen et al. (2023b) illustrate how Violence Against Women Programs actively maintain visibility through social media and local press, coordinating with police and media to ensure those at risk understand their options for seeking shelter [94]. However, the literature also pointed to challenges of communication and media campaigns in including or reaching racialized communities, often resulting in these groups being less likely to seek care, thereby exacerbating health and social inequities [99].

**Theme 3: Capacity to provide and access support services following SLEs.** Following the implementation of public health measures or emergency protocols, such as COVID-19 lockdowns or responses to disasters or climate events, a significant reduction in the capacity of support services inevitably occurred in the aftermath of SLEs. As discussed by Enarson (1999),

Houghton (2010), Fothergill (1999) disasters disrupt the resources used by shelters to help Survivors, including local hotels, safe homes, public transportation, legal/law enforcement, and social/health services [41, 106, 111]. Much of the literature in this scoping review focuses on the COVID-19 pandemic, highlighting how resources were redirected from social and health services to directly combat COVID-19 infection and prevention. This redirection led to the suspension or delay of in-person services and reduced support services, and led to the transition of many services to virtual platforms, with some, like support groups, being canceled due to challenges in maintaining a safe online modality [40, 42, 54, 83]. Consequently, stay-at-home orders and lockdowns not only reduced referrals to support services, limiting survivors' access to assistance, but also posed challenges for both survivors and service providers in accessing in-person sessions during the COVID-19 pandemic. Factors such as physical distancing precautions, decrease in scope of services, limited access to personal protective equipment (PPE), fear of contracting the disease or infecting family members if essential workers, and safety concerns in communal living spaces all contributed to these challenges [39, 42, 47, 48, 52, 58, 60, 62, 68, 72–77, 82, 86, 89, 91, 93–95, 97, 99, 100, 103, 107, 110].

Additionally, reduced screenings in emergency rooms, referrals to support services, and decreased availability of mental and medical health services were reported due to resources being diverted to COVID-19 infection control [49, 55, 78, 85, 91, 93, 94, 101]. As Hendrix (2021) describes, the use of PPE (i.e., masks, goggles, and gowns) during the COVID-19 pandemic posed a significant barrier to making personal connections with patients, leading to a sense of isolation, and negatively impacting the quality of the service provision [49].

Lockdown measures and reduced social support also affected key services, leading to delayed family court proceedings, custody hearings, and extended waiting times for legal processes related to custody, separation, and divorce [37, 47, 60, 88, 93, 101, 102]. These reductions in services and accessibility left individuals compromised, unable to access necessary support, continue therapeutic work, or progress with legal proceedings [93, 97, 102, 107].

Shutdowns decreased in-person services, and remote work arrangements heightened the risk for Survivors, resulting in increased violence and negative mental and physical health outcomes [40, 43, 71, 72, 107]. Physical access to healthcare providers was limited by public health restrictions and physical distancing mandates. This was further complicated by reduced public transportation and thus, impacted Survivors' ability to travel for appointments [42, 49, 51, 52, 60, 63, 66, 74, 75, 102]. Mantler (2024) revealed that inadequate access to essential services, such as transportation in rural areas, significantly hindered the resilience of women attempting to leave abusive relationships, further compounding their isolation in these regions [57].

Clarke (2023) discussed how vulnerable clients were particularly affected by the reduction of services, as the usual avenues for leaving the house, such as grocery shopping, were eliminated due to COVID-19 restrictions [102]. Additionally, childcare became a challenge for many, as isolation measures resulted in children being home due to school closures, while essential workers were required to be at work [61, 112]. Concerns in custody contexts also arose, as there were no safe spaces for exchanging children due to the closure of public spaces [102]. School closures further diminished children's support networks, reducing opportunities for disclosure or confiding in a safe person [90]. Fear of COVID-19 infection also deterred individuals from seeking accommodation in shelters or emergency rooms, with some older women avoiding shelters altogether due to infection fears [49, 51]. The delay or suspension of services further discouraged individuals from seeking help during this period [42] Leat (2024) noted that the unpredictability of public health mandates and policies during the pandemic (i.e., continuously changing) complicated the ability to effectively manage service provision, occupancy, and staffing.

The role of bystanders and informal support networks is crucial in the lives of IPV Survivors. Restrictions, be it emergency protocols or lockdowns, prevented advocates and informal sources of support from accompanying Survivors to appointments or providing emotional support [40, 44, 46, 53, 55, 58, 76, 107].

Authors acknowledged the remarkable commitment, motivation, resilience, creativity, and self-sacrifice demonstrated by service providers and sector leaders during SLEs [39, 41, 80, 86, 89, 100, 104]. Garcia (2022) and Thiara and Roy (2022) discussed how resilience was acknowledged by advocates, communities, and Survivors despite the formidable challenges encountered [46, 99]. Several authors stressed the importance of addressing the needs of support service providers, including provisions for sick leave, flexibility around family responsibilities, flexible working arrangements, training, compensation for extra workload, and support for mental health and well-being [46, 49, 51, 60, 65, 78, 83, 94, 98, 101, 105]. Cunha et al. (2024) highlighted that shelter leadership prioritized the well-being of staff and professionals by regularly checking in and utilizing online platforms for team meetings. This approach allowed for open discussions on professional practices and personal concerns, which fostered a supportive work environment.

**Theme 4: Service providers workload and stress during SLEs.** Across various contexts including the pandemic, economic recessions or climate disasters increased workload for staff in addition to budget cuts, which required them to "do more with less" and take on additional responsibilities (i.e., get involved in homeschooling for children at shelters) [41, 43, 44, 61, 80, 83, 90, 92, 104, 106]. During the pandemic, service provision became more demanding due to ever-changing public health measures like disinfection protocols and PPE requirements, combined with a reduced workforce, all of which contributed to pandemic fatigue. [51, 56, 60, 65, 68, 71, 91, 99, 101–103, 105]. The majority of literature (94%) in this scoping review was pandemic-focused, and delved into the constantly evolving public policy responses, leading to profound exhaustion and burnout within the VAW sector. This exhaustion significantly impacted the stability and sustainability of service providers and their efficacy in delivering support.

Service providers talked about the mental and emotional toll of service provision during the pandemic, and reported increased anxiety, stress, depression, and feelings of isolation for themselves, alongside concerns for their clients and children [39, 42, 65, 82, 93, 94, 100–102, 107]. Additionally, providers were overworked, overloaded, and became sick and stressed, and needed to balance virtual shelter operations with childcare responsibilities while home-schooling [52, 56, 59–61, 69, 83, 89, 95, 99–102, 109]. Service providers, particularly in rural areas, faced additional stress due to limited resources and the challenges of conducting secure virtual sessions, especially when Survivors were isolated with their abusers [47, 97]. As Voth Schrag (2023) discussed, ensuring technology safety became an integral aspect of the job for virtual service providers, adding complexity to the delivery of safety plans as virtual service provision evolved [73]. This often meant prioritizing urgent matters, such as immediate service provision, at the expense of advocacy and group sessions [51]. Moreover, lack of mentorship and institutional support and difficulties in contacting their own organizations/colleague networks for assistance or guidance during crises were reported, shifting emotional support from addressing Survivors' trauma to dealing with pandemic-related challenges and virtual work, which necessitated additional training [42, 60, 93, 94, 100, 110]. Front-line staff, who typically face more challenging working conditions with fewer resources and lower pay, felt a pronounced disconnect with top management, who were often physically separated from the shelters and seemingly unaware of the daily risks and exposures frontline staff endured [37]. Of note, Burd (2023) found that the implementation of public health restrictions during the COVID-19 pandemic challenged *the principle of choice*, leading to the disempowerment of

women and contradicting the feminist principles that underpin their work. Staff struggled with prioritizing COVID-19 public health mandates over providing their core VAW services.

Several authors highlighted the significance of funding and financial support [37, 39, 44, 61, 65, 87, 95, 105, 106]. They emphasized the necessity for adaptable sustainable funding mechanisms, especially when adjusting service delivery in response to SLEs. In the context of the COVID-19 pandemic, funding flexibility was required to transition to virtual modalities or invest in innovative strategies to reach isolated Survivors. This flexibility is crucial for creative and adaptable services that effectively address the changing severity and fluctuations in IPV case numbers amidst the dynamic and continuously changing characteristics of SLEs [37, 65, 86, 90, 91, 93, 100, 103].

**Theme 5: Issues of isolation, power dynamics and control during SLEs for IPV survivors.** Amidst the unprecedented challenges precipitated by SLEs, both IPV Survivors and service providers found themselves grappling with isolation, exacerbating existing vulnerabilities, and complicating access to vital formal and informal support services. As Enarson (1999) describes, "subject to a vicious cycle of power and control, battered women live in a world of increasingly narrow social networks with abusers who keep them isolated, restrict their transportation and employment opportunities, and control household resources" (p. 748). Women living in daily fear and intimidation and who experience emotional crises before a disaster already face enough hardship, and when compounded with having to navigate evacuation or lockdown warnings, face even more difficulty [41].

Survivors experienced isolation from family, friends, and broader social networks [109]. Pandemic lockdown mandates left many Survivors isolated and disconnected from their usual informal support when at home with their abusers [39, 45, 47, 52, 58, 67, 71, 75–78, 82, 83, 91, 97, 99, 102, 108, 110]. Numerous articles delved into the concept of social isolation [15, 58, 68, 72, 74, 109]. For example, perpetrators of violence employed deliberate tactics to cut off Survivors from external support networks, rendering them without access to assistance or resources, and heightened their dependency on the abusive partner with decreasing motivation to seek help. This led to an increased risk of experiencing further violence. Notably, Gregory & Williamson (2021), Holt et al. (2022), and McKinley et al. (2023) investigated the isolation felt by informal support providers, who were overwhelmed themselves and depended on their networks to assist the survivors in their lives [89–91]. Similarly, service providers felt isolated from their work networks and community during the pandemic [60].

Survivors encountered increased isolation from formal services due to reduced, suspended, unavailable, or virtualized services [57, 91, 101, 105]. Many found themselves unable to access or engage with services virtually due to their confinement at home with their perpetrator, as there was a heightened risk of violence if the Survivor was discovered seeking support. The presence of other family members also hindered Survivors' ability to seek help [43, 46, 51, 53, 57, 63, 66, 67, 74, 79, 95, 104]. Furthermore, the COVID-19 lockdowns prolonged the cycle of violence by confining Survivors with their abusers and eliminating opportunities for respite or safety planning, such as commuting to work or running errands [53, 87]. Concerns about custody arrangements during the pandemic and fear of retaliation by the abuser further deterred Survivors from seeking help or leaving abusive situations. Those who accessed shelters were isolated upon entrance to the facility due to lockdown measures [73].

The pandemic-induced isolation, enforced through lockdown mandates, served as a tool for perpetrators of IPV to assert control and manipulate Survivors through power dynamics. Several articles delineated instances of "exerting power and control" and "weaponizing the pandemic" to restrict movements, coerce Survivors into cohabitation, or instill fear about the virus, alongside instances of financial abuse and withholding essential resources [40, 45, 48, 51, 53, 58, 63, 74, 77, 78, 84, 86–88, 90, 91, 93, 95, 99, 100, 109]. Examples include preventing them

from going to work, preventing child contact with mother in custody cases, using food pensions, limiting access to PPE, and limiting access to social medical appointments (i.e., hiding insurance cards).

Moreover, the isolation mandates inadvertently re-traumatized IPV Survivors by mirroring control and manipulation tactics employed by their perpetrators, exacerbating feelings of powerlessness and entrapment within a confined environment [39, 40, 48, 63, 66, 72, 73]. Survivors detailed how wearing masks brought on feelings of being suffocated, strangled, and sexually abused with obstructed breath [63, 73]. Others discussed being "retraumatized" by the pandemic lockdown restrictions and when restrictions were implemented at the shelters, it mimicked the situation of being isolated and restricted by a controlling and abusive partner (i.e., no communal/congregate living, isolation and quarantine in the rooms/homes, controlling of movements, and enforcing infectious disease control and safety precautions) [40, 48, 57, 63, 66]. Toccalino et al. (2022) described how continuously changing physicians during the pandemic meant Survivors repeated their stories, resulting in re-victimization, which was an aversion to seeking help [72].

To assist Survivors in navigating various systems and fostering resilience, peer support groups, domestic violence advocates, liaison roles and system navigators emerged as necessary components [45, 49, 72]. In some cases, the pandemic-induced isolation provided an unexpected opportunity for introspection and healing, with Survivors finding refuge in shelters that facilitated a focus on personal safety and learning about IPV tactics [66]. Additionally, the public health restrictions (i.e., visitor policies in healthcare settings) facilitated disclosures of abuse to healthcare providers when in person, as perpetrators were unable to accompany Survivors to medical appointments, allowing for a safer space for disclosure [49, 102]. Due to restrictions, youth IPV Survivors, experienced a sense of safety due to lessened encounters with perpetrators whom they did not live with, highlighting the unintended positive outcomes of pandemic-induced isolation [100].

To maintain personal connections and support during the isolating lockdown pandemic periods, staff implemented safety measures beyond the use of PPE, including holding meetings in open, non-traditional spaces such as parking lots [39]. In some instances, shelters provided transportation services to facilitate safe movements during lockdowns [52]. Additionally, maintaining connections with informal support systems (i.e., support bubbles of family and friends) was identified as critical in supporting survivors with emotional and practical support during challenging times [56, 67, 91].

**Theme 6: Navigating the transition to virtual service amidst the pandemic: Challenges and Innovations.** The literature on transitioning to virtual service provision was only in the context of the pandemic; the papers on economic recessions and natural disasters did not include virtual support services. Navigating the transition to virtual platforms for formal IPV support services amidst the COVID-19 pandemic presented several challenges, disrupting traditional modes of assistance and requiring innovative approaches to ensure continued care for Survivors.

Several authors discussed the issue of the digital divide, exacerbating disparities in accessing virtual care for IPV Survivors during the pandemic, underscoring the unequal distribution of technological resources (i.e., ability to afford or have digital platforms) and skills (i.e., older adults) reducing their ability to seek or access support online [39, 40, 48, 60, 67–69, 71, 81–83, 88, 91, 93, 94, 97, 98, 100, 101, 104, 109, 110]. The transition also emphasized service providers' lack of experience and training in delivering telecare, along with issues such as access to devices and reliable connectivity, posing significant hurdles in providing effective support [42, 46, 58, 59, 65, 69, 72–74, 89, 90, 93, 94, 101–105, 109]. Concerns regarding the sustainability of technology emerged, with authors noting service providers (and their clients) were

experiencing screen fatigue. They found it challenging to complete complex online forms, highlighting the limited capacity of both Survivors and service providers to navigate online systems beyond care provision [59, 61, 63, 69, 73, 75, 90, 101, 104].

Technology emerged as a double-edged sword. On one hand, it provided a venue for accessibility, on the other, it was evident that perpetrators increasingly weaponized it for surveillance, digital violence, and monitoring purposes during the pandemic, exacerbating fears of privacy invasion and further threatening survivor safety [61, 81, 82, 84, 108, 109]. The transition to virtual and telephone platforms also raised concerns about the efficacy of risk assessment and remote counseling with some Survivors and service providers expressing difficulties in establishing trust, rapport, interpreting visual cues and emotional connections, crucial for therapeutic relationships, healing trauma, and IPV disclosure [39, 40, 46, 48, 59, 60, 65, 67–69, 73, 74, 79, 82, 84, 86, 89–91, 93, 94, 99, 101, 104, 105, 110]. In post-separation contexts, Desai (2022) explained how perpetrators of abuse adaptively and persistently use technology-facilitated methods to stalk and harass their former partners, leading to heightened anxiety about safety for the Survivors and their children.

Innovative strategies emerged to enhance services through virtual modalities, ensuring survivor safety. The utilization and positive impact of virtual platforms have been explored and implemented in various studies [37, 39, 40, 51, 53, 55, 60, 61, 65, 67, 71, 75, 82, 83, 85, 93, 94, 98–101, 105, 107, 109, 110]. These studies underscored the significance of expanding broadband internet services to ensure seamless connectivity [62, 69, 103] and highlighted the advantages of virtual meetings for support staff including reducing the need for physical transport, minimizing exposure risks, and saving time [60, 109].

Fogarty et al. (2022) discussed the favorable aspects of online telehealth, while Golenko & Rittossa (2022) described the increased usage of online and telephonic communication for support services [88, 105]. Additionally, discreet digital communication platforms observed or proposed [53, 76, 104], alongside initiatives to counter digital exclusion by providing devices and internet access to Survivors and service providers [94, 103]. Oswald et al. (2022) proposed that broadening access to broadband internet services would not only enhance telehealth and IPV resource accessibility but also help IPV survivors preserve essential social connections [62]. Gregory (2021) touched on the resourcefulness of informal support providers in communicating with Survivors to offer assistance [51, 89]. According to Bomsta and Kerr (2023), telehealth proved not just a substitute but an enhancement over in-person services, offering clients significant benefits that reduced the need for travel and related expenses, enhanced privacy, and convenience. This led to fewer missed appointments and high attendance in support groups conducted via telehealth [37]. Similarly, as Desai et al. (2022) described, some Survivors felt more comfortable and safer engaging in telephone counseling as it reduced self-consciousness and provided a sense of security and privacy [84].

Fogarty (2022) emphasized the need for adaptability, particularly in tailoring therapeutic approaches to help effectively engage with clients online [105]. This involves establishing rapport with women and devising innovative methods to involve children in the therapeutic process [110]. Coram (2021) explored innovative strategies to sustain client engagement, such as offering footpath drop-offs and food parcels as part of service provisions [103]. Some studies recommended extending helpline hours to accommodate the times when clients most need support and suggested that proactive service delivery in convenient off-site locations, rather than traditional drop-in centers [79, 99].

Leigh (2022) recommended leveraging local media, virtual town halls, and live feeds on Facebook to raise awareness about ongoing services [53]. Yakobovich et al. (2023) recommended the implementation of educational curricula and public awareness campaigns focused on healthy relationships, gender expression, and the dynamics of violence, with targeted

outreach to educate men and boys on issues of gender-based violence [78]. In the context of perinatal health, Krishnaumuti (2021) developed a mobile application with an IPV assessment tool for pregnant women, providing a means to reach individuals hesitant to disclose directly to their care provider [50]. Additionally, Rahman (2022) emphasized the necessity of mandatory pediatric primary care visits, irrespective of emergency orders, as an avenue for Survivors to access support services [64].

Overall, the transition to and utilization of virtual services underscored both the resilience of Survivors and an imperative for ongoing innovation and adaptation to meet the evolving challenges of IPV support amidst the pandemic, or any SLEs. As highlighted by Clarke (2023), this opportunity compelled certain organizations to carefully evaluate their operational procedures, fostering enhancements and prompting fresh perspectives on virtual administrative work, as well as the optimization of services for Survivors and their children in shelters [102].

**Theme 7: Privacy, confidentiality, and safety.**   The articles discussing safety concerns for Survivors of IPV were mostly in the context of the COVID-19 pandemic, particularly regarding confidentiality breaches, diminished physical safety, and challenges in safety planning within the context of heightened isolation and restricted resources. Enarson (1999) shed light on "second-order evacuation" in the context of environmental disasters requiring women to evacuate from their safe spaces (i.e., shelter, motel, family) and the inability of evacuation centres to protect privacy or ensure Survivor's safety, especially in small communities [41].

Survivors faced reduced privacy while accessing services virtually or disclosing experiences of violence in person due to restrictions and the presence of perpetrators or family members in the home, leading to heightened safety concerns during pandemic lockdowns [39, 42, 46, 47, 51, 60, 69, 71, 72, 74, 75, 79, 81, 82, 93, 94, 97, 98, 102, 104, 105, 109, 110]. In some cases, small communities (i.e., rural, remote) reinforced privacy and confidentiality concerns among Survivors, deterring them from seeking or accessing help due to fears (i.e., stigma, shame) of their situations becoming public knowledge in their close community [54, 57, 93].

Similarly, service providers grappled with privacy and confidentiality issues in their own homes while providing services, navigating difficult stories with family members present as they transitioned to work at home during the pandemic [46, 47, 61, 62, 68, 81, 90, 104, 108]. Aligning public health measures with survivor and shelter confidentiality presented challenges, particularly regarding reporting positive cases while upholding confidentiality [61]. Additionally, interpretations of healthcare policy and legislation, particularly regarding data security and privacy, hindered the implementation of virtual care within organizations, with current privacy laws failing to adequately address the nuanced understanding of "privacy" and "safety" within the virtual environment [59]. Stanley et al. (2022) revealed the challenges of delivering confidential and safe services through community touchpoints in rural areas with small populations [98].

Navigating safety planning—plan that identifies methods to enhance the safety of victims and their children when confronted by family violence [113]—and accessing assistance for IPV Survivors became increasingly complex during the pandemic, necessitating innovative approaches to ensure protection within the constraints of social distancing measures and limited resources. Challenges included providing support for Survivors attempting to escape violence or safety plan amidst lockdowns, where finding excuses to leave without raising suspicion was difficult [40, 58, 61, 71, 89, 90, 108, 109]. Informal supports, such as bystanders, neighbors, friends, and family members, hesitated to intervene due to fear of reprisals or negative consequences, further isolating Survivors [96].

Measures to ensure security and privacy in virtual settings were prioritized through the adoption of encrypted web-based video call services [109]. Ghidei et al. (2022) and Ragavan et al. (2022) explored safety measures, including the incorporation of safe words to conclude

virtual sessions or the use of code words via text messaging [47, 63]. Furthermore, Cortis (2022) emphasized the criticality of adjusting safety protocols to changing circumstances, such as conducting screenings in virtual environments when children were present [104].

Stanley et al. (2022) explored how banks, pharmacies, and online platforms served as '*touch-points*,' which are *safe spaces* providing discreet venues where survivors could seek assistance [98]. Additionally, several studies also highlighted various strategies of IPV Survivors using code words and phrases to seek supports discreetly. These tactics are vital for allowing Survivors to seek help without alerting their abuser [71, 98, 99, 110]. Pharmacies utilize the "Ask for ANI" code word system, providing survivors with a discreet method to signal their need for help [98]. Additionally, organizations have developed systems using words such as 'sunshine' to indicate safety and 'cloudy' to signal danger, enabling women to discreetly communicate their status and safely plan escapes [99]. Castellano-Torres et al. (2023) introduced the "mask 19" initiative from Spain during the COVID-19 lockdown, which allowed women at risk to discreetly request help at pharmacies using a code word, offering a confidential way to access support without alerting their abuser or attracting public attention. This initiative was part of broader efforts to ensure continued access to support services during the pandemic, catering to the heightened need for discreet assistance amidst increased isolation and the potential stigma or danger of discussing such issues publicly. [82].

**Theme 8: Intersectionality and issues faced by racialized and marginalized communities.**   The concept of intersectionality emerged in several of the papers, highlighting the disparities and barriers which intersect with race, ethnicity, substance use, mental health, and socioeconomic factors. This intersectionality often translates into greater challenges for marginalized and racialized communities in accessing support and resources for Survivors of IPV.

In the context of the climate emergency, Brown et la. (2010) highlighted the increasing number of women seeking shelters who grapple with multiple serious issues such as substance use, mental health concerns, and homelessness [38]. Moreover, austerity measures resulting from economic crises, as discussed by Briones-Vozmediano et al. (2014), exacerbate the structural violence experienced by immigrant women, limiting their employment opportunities, and impeding their ability to leave abusive partners [80]. These authors further explain how women without legal status or valid documentation often face additional barriers such as language barriers and lack of social support networks, further limiting their access to support services and resources Several papers have underscored the impact of the COVID-19 pandemic on oppressed, immigrant, migrant, and refugee communities, exacerbating existing challenges in accessing services (i.e., legal or social services assistance in applications following job loss). These challenges are compounded by factors such as language barriers, cultural norms about disclosure, lack of culturally safe care, food insecurity, age, and geographic isolation (i.e., rural, remote), leading to disparities in service utilization [46, 47, 57, 60, 67, 68, 72, 84, 98, 99]. Fuchsel (2024) found that perpetrators threaten immigrant women from seeking help, which leads to fears of deportation, uncertain immigration status, limited language proficiency, and cultural stigma/shame, thereby exacerbating their vulnerability and isolation during the pandemic [45]. In addition to the various inequities already faced by racialized and marginalized communities, Safar et al. (2023) highlighted a specific gap in the system: a lack of appropriate supports for older women, who often remain overlooked within these groups (63).

Additionally, pre-existing resource deficiencies for adults with disabilities, older women, youth, rural populations, and culturally diverse individuals were exacerbated during the pandemic and in the context of climate disasters [44, 46, 47, 67, 78, 94, 98, 99]. In their study, Shyrokonis et al. (2023) revealed that racialized individuals, essential workers, and those in financially intertwined marriages face heightened risks of inequities due to a lower likelihood of seeking support. Conversely, those who are young to middle-aged and pregnant are more

likely to seek support [70]. Castellanos-Torres (2023) reported an increase in demand among women facing various forms of social vulnerability, such as unemployment, job insecurity, or belonging to an ethnic minority, factors that have facilitated the perpetration of sexual violence. Randell et al. (2023) noted that staff from marginalized communities encountered challenges arising from the intersection of COVID-19 and racism, affecting both their personal and professional lives [65].

Ghidei et al. (2022) highlighted how structural inequities compounded with IPV and the COVID-19 pandemic created unique challenges for marginalized groups of IPV Survivors [47]. More specifically, several of the papers investigated the structural inequities and barriers faced by racialized and immigrant populations experiencing IPV, along with associated intersecting issues such as substance use, homelessness and mental health. These barriers include xenophobia, racism, homophobia, transphobia and other forms of discrimination [43, 45, 49, 56, 63, 78, 91, 110]. Toccalino and colleagues (2022) emphasized the necessity of race- and culture-based data to demonstrate the structural inequities encountered by oppressed populations [72].

Anitha and Gill (2022) draw attention to the often-overlooked experiences of racially minoritized women during the COVID-19 pandemic, from the perspective of frontline practitioners of the same communities. The study showed how frontline workers adjusted their methods and used their authority to tackle the specific challenges and inequalities faced by racially minoritized women, highlighting important links between race, gender, and organizational actions. Similar findings from Thiara and Roy (2022) emphasized that racialized organizations serving racialized Survivors begin at a disadvantage, lacking initial resources and facing competition with larger mainstream organizations. They also confront systemic barriers while striving to provide culturally and racially relevant services [99].

From the perspectives of Survivors, Gill and Anitha (2023) discussed how racialized Survivors navigate the intersection of gendered and racialized disadvantages, which influence both the nature and extent of the abuse they endure and their opportunities for seeking formal and informal support. This is further complicated by a historical mistrust between police and racialized communities that often discourages them from seeking help [87]. Similarly, Luebke (2023) described how Indigenous Survivors often face compounded barriers due to distrust and discrimination, which manifest in their experiences of not being believed, listened to, or taken seriously within law enforcement and healthcare systems. These interactions, marked by racism and sexism, underscore the intersectional challenges of being Indigenous, female, and low-income, further exacerbated by a lack of culturally relevant care in the structures meant to provide protection and health services [54].

Numerous authors highlighted the significance of delivering community-based, trauma-based, trustworthy, language sensitive, and culturally safe supports to high-risk sub-groups of IPV Survivors [42, 45–47, 54, 72, 78, 79, 93], with some advocating for or implementing a harm reduction approach [39, 63]. Several authors discussed how service providers navigate the tension between adhering to public health mandates and understanding the unique risks to survivors in their communities. They emphasized the advocacy roles of these providers in mediating and influencing policies for more funding and to address structural inequalities affecting minoritized women, thereby enhancing their ability to provide culturally and contextually appropriate support [39, 65, 79, 87].

**Theme 9: Financial strains and abuse.** Financial strains resulting from various crises—whether it be a pandemic, economic recession, or climate disaster—contribute significantly to the increase in IPV. The dimension of financial abuse during SLEs is crucial in empowering women, particularly in enabling them to leave abusive relationships. Vives Cases et al. (2021), Safar et al. (2023) and Randell et al. (2023) specified the need for financial support for

Survivors [65, 67, 100]. Financial independence and access to resources are pivotal factors in the ability to leave without being financially dependent on the abuser. Several articles have highlighted that economic precariousness during any SLE makes it less likely for Survivors to leave abusive situations (e.g., loss of job), access services (e.g., due to associated costs, lack of health insurance, perpetrator control of personal documentation), and if financially dependent on the perpetrator of violence, they are also trapped in hostile environments which exacerbate IPV (stress, frustration) while decreasing their ability to seek support [40, 42–45, 58, 60, 63, 65, 72, 74, 75, 92, 95, 106, 107]. Housing insecurity exacerbates the risk of IPV, highlighting the complex interplay between socioeconomic instability and the heightened risk of remaining in abusive relationships. This insecurity, along with deteriorating living conditions, often forces women to endure abusive and stressful situations [37, 38, 53, 54, 56, 74, 78, 80, 86, 94] or rely on emergency shelters or insecure hotels for extended stays due to challenges in transitioning to housing [65, 83]. Healthcare professionals expressed concerns about discharging Survivors —who presented at emergency departments with severe injuries—to hotels, due to the lack of safe and secure community resources during COVID-19 [49].

Supplement 5 (S3 Table) presents the themes from the analyzed articles, detailing the barriers and facilitators identified in the literature for each of the included articles. To interpret Supplement 5, review the themes column, which synthesizes barriers and facilitators from the analyzed articles. Each row corresponds to an individual article, outlining the specific challenges and supports identified in the literature regarding access to services for IPV survivors during stressful life events. Figs 2 and 3 are an iteration of this supplement to provide a visual of the key themes and corresponding authors for barriers and facilitators, respectively.

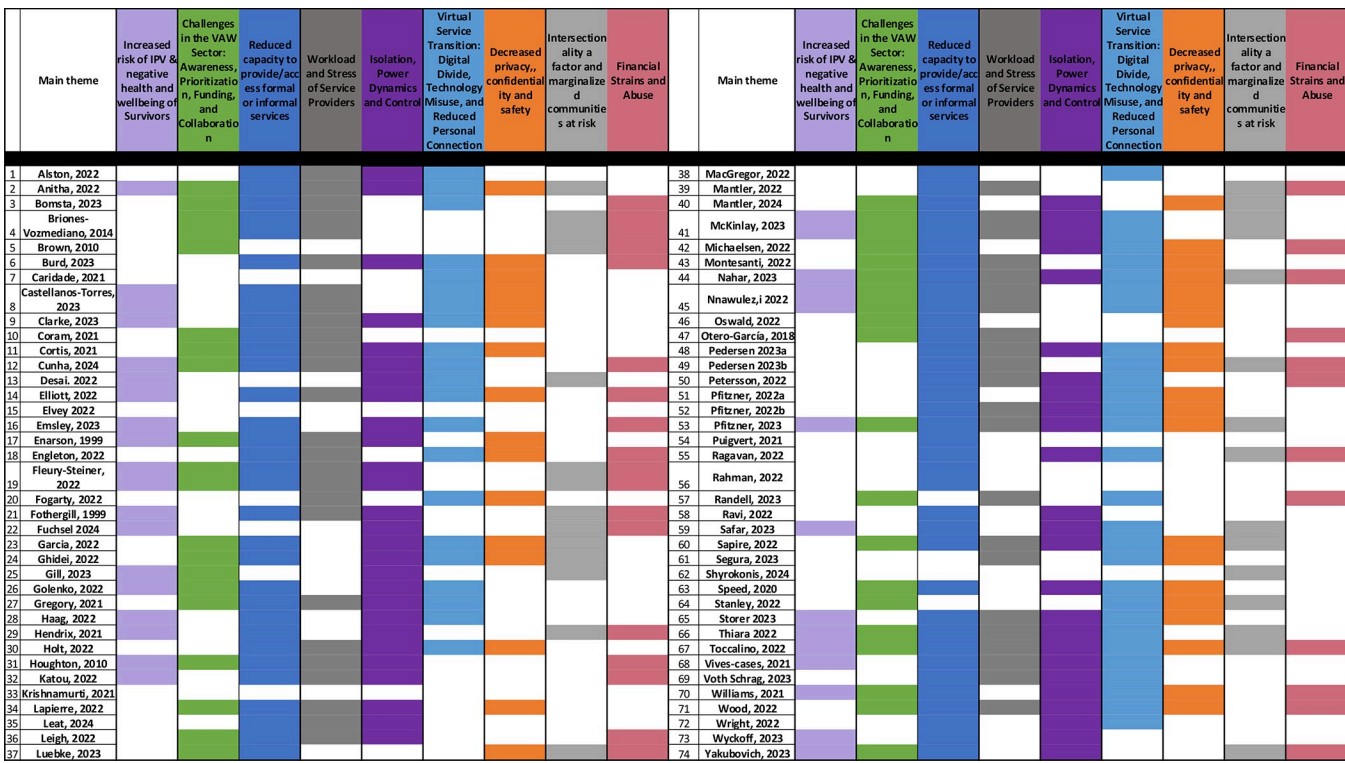

**Fig 2. Barriers to accessing IPV supports during SLEs.**

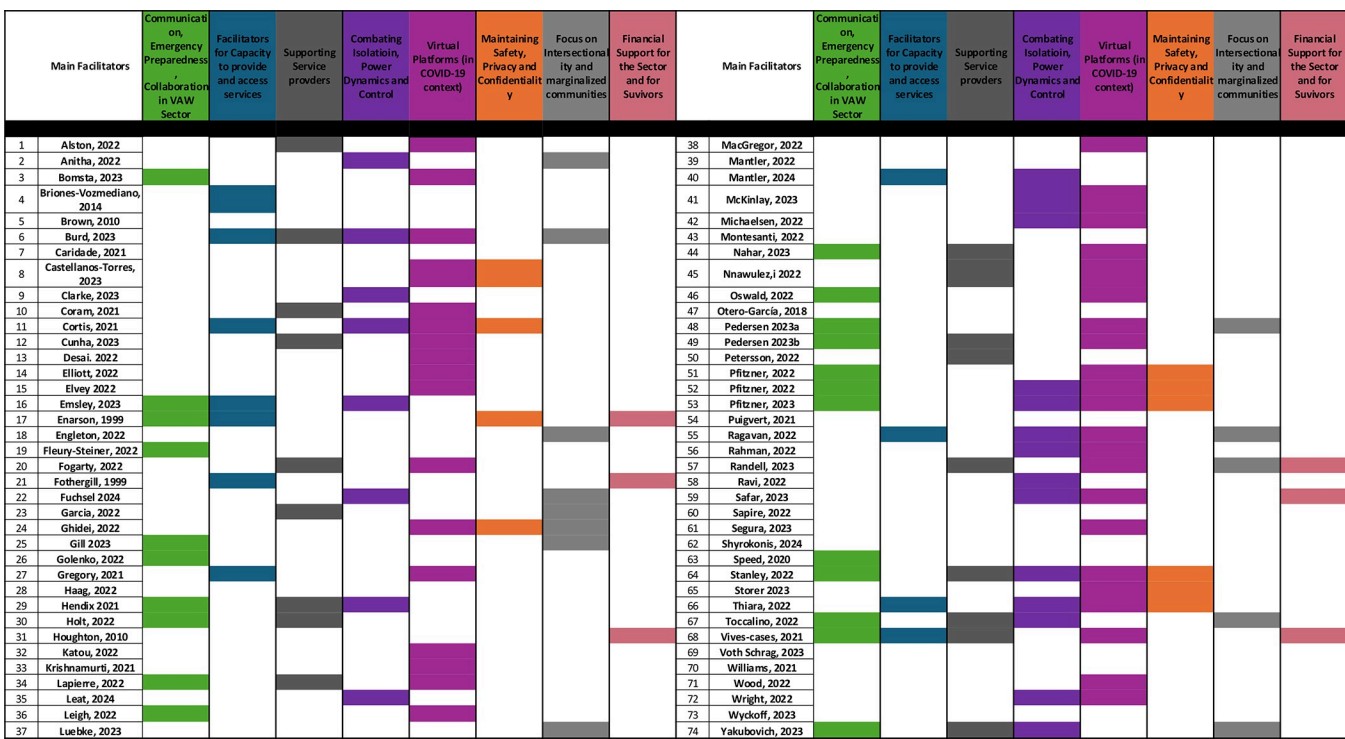

**Fig 3. Facilitators to accessing IPV supports during SLEs.**

## Discussion

This scoping review examined literature about access to services for individuals experiencing IPV during SLEs, focusing on pandemics, natural disasters, and economic recessions. Nine key themes were identified, including, the impacts of SLEs on IPV survivors' health and well-being, considerations for the violence against women (VAW) sector, capacity to provide and access support services, service providers' workload and stress, issues of isolation and power dynamics, the transition to virtual services during the pandemic, privacy and confidentiality concerns, intersectionality and challenges faced by marginalized communities, and financial strain and abuse. The findings highlight barriers and facilitators for accessing services by Survivors and their social and health support systems—both formal (e.g., social and health service providers) and informal (e.g., family, friends, neighbors)—during and after SLEs.

### Context of stressful life events (SLEs)

Most of the studies in this review focus on the COVID-19 pandemic, highlighting a significant gap in research concerning economic recessions and environmental disasters. Before the pandemic, research focusing on access and provision of services for IPV Survivors was limited, with much of the literature primarily addressing increased rates and likelihood of IPV during SLEs [14, 114]. Studies dating back to the Great Depression discussed the impacts of economic uncertainty on marital relationships [10, 11, 13]. Research examining the 2008 Great Recession in the U.S. suggested a positive association between household-level unemployment and economic hardships with abusive behavior in a relationship [13]. Furthermore, Schneider et al. (2016) demonstrated a correlation between rapid increases in unemployment rates and heightened male partner controlling behavior, even after adjusting for unemployment and economic stress, indicating that anticipated anxiety and uncertainty negatively impact relationships [13].

Since Elaine Enarson (1999) explored domestic violence interventions in North America post-disasters, there has been limited research addressing service access and provision for Survivors amid environmental and natural disasters [41]. Similarly, studies related to access and provision of formal (i.e., health and social services) or informal (i.e., family, friends, neighbors) supports in economic recessions are scarce, as only a few papers identified this in our review. A small study conducted post-2008 economic downturn in Ireland suggested that the recession directly impacted VAW services and their funding [115]. However, details for its inclusion and extraction are unavailable as the full article was not accessible, and the authors did not respond to inquiries.

As a result, much of the work in high-income countries has focused on the severity or change in experience of violence during SLEs, with limited emphasis on accessibility or provision of services until the emergence of the COVID-19 pandemic. The urgency to address the global pandemic, and the subsequent "shadow pandemic" of IPV, has spurred increased rapid reviews and studies regarding accessibility to formal and informal IPV supports. Consequently, advocates are calling for commitments to evidence-based, practice-informed violence prevention in the VAW sector (Accelerator for GBV Prevention, 2024), with the engagement and inclusion of lived experiences of IPV Survivors at the community level. Amidst increasing global extreme weather events as well as the economic volatility—exacerbated by the ongoing effects of COVID-19 and persistent conflicts in regions like Ukraine and Middle East—there is an imperative for governments to prioritize the provision of services to IPV Survivors within these SLE contexts.

## Participant perspectives–Survivors of IPV and support providers

In terms of participant perspectives, about 76% represented formal service providers or a combination of Survivors and service providers, 23% represented Survivors, and a mere 1% from informal supports (family, friends, neighbors, bystanders). A notable gap persists in the literature regarding the lived experiences of both IPV Survivors and informal support networks.

Understanding the experiences of IPV Survivors during SLEs is necessary to respond through services and design policies. Accordingly, their voices often go unheard due to isolation, stigma, safety concerns, re-traumatization, and distrust of research, leading to underrepresentation in relevant studies and exacerbation of health and social inequities [116, 117]. To address this at the organizational and community level, more research should involve IPV Survivors to ensure their perspectives inform policy and programming decisions. Such efforts must adhere to rigorous ethical considerations to minimize harm [118], with community-based participatory research and engaged scholarship emerging as promising approaches in the realm of IPV [119–123].

At the policy level, when formulating policies, programs, and treatments to ensure services are appropriate and accessible for Survivors of IPV, there should be targeted efforts toward marginalized, oppressed, and at-risk groups (e.g., racialized communities, immigrants, refugees, gender diverse persons, persons with disabilities). These groups face intersecting oppressions that necessitate nuanced and culturally informed approaches [112]. Additionally, consideration of social determinants of health is integral, as IPV Survivors often require comprehensive support including housing, financial assistance, social support, employment, and safety planning during periods of isolation. Wrap-around services, including counseling, employment assistance, housing support, vocational training, and resume writing, are crucial [124]. Policymakers must also consider the unintended consequences of mandates (e.g., stay-at-home orders) on at-risk communities such as IPV Survivors, who may face increased isolation and reduced access to services when confined with their perpetrators.

Furthermore, there are implications for research focusing on the lived experiences of informal support providers—whether family, friends, bystanders, or neighbours. Informal social supports significantly contribute to the well-being of IPV Survivors, offering both emotional and tangible assistance that can mitigate the adverse psychological effects of abuse and reduce susceptibility to further harm [16]. Moreover, these supports are associated with improved health outcomes for IPV Survivors, highlighting the importance of community-based interventions and advocacy services in enhancing access to resources and coping strategies [17, 120].

## Emergency management and preparedness planning

This review revealed a significant oversight; IPV is often sidelined in emergency response and readiness protocols, despite well-documented evidence of increased IPV and domestic violence prevalence during and post SLEs. The lack of consideration for IPV in emergency preparedness at the policy and organizational levels has left Survivors without adequate support, exacerbating their isolation and vulnerability. At the macro level, high-income countries may have national action plans to address gender-based violence (GBV) or emergency plans that consider social vulnerabilities, including Survivors of IPV. However, it is noteworthy that some countries have only just begun addressing the impact of disasters on the rise of GBV, particularly in the aftermath of the COVID-19 pandemic [125–127]. At the organizational or community level, crises often trigger a "survival" mode, compounded by a lack of emergency preparedness within organizations, leading to confusion, provider burnout, and challenges in reaching clients.

There are significant implications for emergency preparedness plans at this level, necessitating coordination and partnership among key actors within the IPV service provider community. A coordinated response among various organizations during emergencies, including local health organizations, medical professionals, support services, and community-based organizations, is essential to ensure necessary services are readily available. Furthermore, in the absence of formal policies or plans, community resilience and self-organization tend to emerge during SLEs and become critical at the community and individual levels. It is important to amplify the voices of service providers and IPV Survivors and ensure thay are reinforced in emergency management planning and integrated into response and recovery plans [128, 129].

IPV is fundamentally linked to gender disparities, disproportionately affecting women and girls [130, 131]. The COVID-19 pandemic highlighted these issues through the implementation of stay-at-home mandates, which inadvertently heightened IPV risks by confining survivors with their abusers and disrupting support services—predominantly staffed by women—thus exacerbating health and social inequities among women. Despite these challenges, national emergency management plans frequently overlook gender-specific considerations. Recent Canadian initiatives emphasize the necessity of addressing the gendered impacts of disasters, including surges in gender-based violence, by fostering awareness and adopting already established guidelines [125, 132]. Future strategies must reassess emergency management plans to incorporate gender as a consideration, recognizing the unique impacts on different genders and ensuring that emergency responses include definitive measures to maintain support services for IPV survivors, prioritizing access for individuals who are disproportionately affected.

## Financial and funding consideration for the sector

Regrettably, government funding at the macro level tends to be reactive, where it is primarily allocated for emergency response purposes, displaying conservatism in funding social support

programs during times of crisis. This austere reaction poses significant challenges to sustainability and long-term planning efforts. Moreover, funding and resource allocation decisions often fluctuate based on political agendas, lacking consistency and hindering reform efforts. Policymakers need to develop more resilient funding mechanisms that ensure consistent and adequate support for the sector, particularly during times of crisis. Additionally, when targeted funding is allocated for an emergency response, it will be important to ensure adaptive and flexible funding mechanisms that can quickly respond to emerging crises.

## IPV—A human rights and social justice lens

In the articles reviewed, a consistent theme emerged regarding the adverse effects of SLEs on the health and well-being of IPV Survivors [7, 133, 134]. This includes increased severity and frequency of abuse, social isolation, and limited access to support networks. It also shows an exacerbation of pre-existing trauma and hinders access to healthcare and support services, particularly for racialized communities.

The pervasive adverse effects of SLEs on IPV Survivors underscore a broader human rights and social justice issue, where systemic barriers not only perpetuate the cycle of violence but also impede the fundamental rights to safety and well-being [135, 136]. Addressing these issues is imperative from a health equity perspective as it is a societal moral obligation to uphold Survivor's human rights to life, health, personal freedom and security, free from violence [136]. Additionally, IPV represents a profound social justice issue as it reflects and reinforces inequalities across gender, socioeconomic, and racial lines, which in history, have led to Survivor disadvantage across several social and economic conditions, affecting their health and well-being [137]. By addressing IPV through concerted social change and advocacy, we recognize IPV not as a personal tragedy, but rather a societal failure that requires systemic interventions to ensure justice and protection for all affected, and especially higher risk populations facing health and social inequities.

The United Nations (UN) 2030 Agenda for Sustainable Development provides a comprehensive framework for addressing complex global issues—such as IPV—underscoring the interconnectedness of health, gender equality, and social equity through several sustainable development goals (SDGs) and an urgent call to action in global partnership [138]. IPV is a complex issue that requires a multidimensional response that aligns with several key UN SDGs, particularly SDG 3 (Good Health and Well-being), SDG 5 (Gender Equality), and SDG 10 (Reduced Inequalities). *Good Health and Well-being* aims to ensure healthy lives and promote well-being at all ages, which is crucial in the context of IPV where survivors often face serious health consequences, both physical and psychological. Addressing IPV helps mitigate these health impacts and supports broader public health goals. *Gender Equality* is central to combating IPV, as it predominantly affects women and girls. Strengthening efforts to achieve gender equality helps reduce the prevalence of IPV and supports survivors in accessing necessary services and justice. *Reduced Inequalities* emphasizes the need to address inequalities that exacerbate the risks and effects of IPV for marginalized groups—such as racial minorities, immigrants, and individuals in low socio-economic settings—who face compounded challenges in accessing support and resources. Policies, programs and approaches that address these key SDGs contribute to a global partnership to reducing IPV and making the 2030 Agenda a reality.

## Key implications for research, interventions and policy

Future research should explore access and provision of services for IPV Survivors across various SLEs, incorporating perspectives from Survivors and informal support networks.

Additionally, addressing social determinants of health and nuances within marginalized communities is crucial for informing evidence-informed interventions. There is a call for using an intersectional social justice approach to intervention and policy development in the IPV sector [137, 139].

Policies should prioritize the needs of IPV Survivors, ensuring that adequate support and resources are available to address the complexities of IPV within SLE contexts. Policymakers and service providers must target marginalized and at-risk groups while considering social determinants of health to ensure services are appropriate and accessible. Government funding for IPV services must be consistent and sustainable, with a focus on long-term planning and core funding [140] rather than following reactionary responses during emergencies. Additionally, emergency preparedness planning and policies should incorporate IPV at all levels to mitigate the inequitable impacts of crises on IPV Survivors attempting to access social and health supports [125]. Virtual services during COVID-19 presented both challenges and opportunities for IPV Survivors' accessing care. While the shift to virtual platforms allowed for continued support during isolation, it also highlighted some key barriers, particularly the digital divide. To effectively address these disparities, policy should focus on ensuring equitable access to broadband internet services and necessary devices [62, 69, 94, 103], enabling all individuals to benefit from virtual care if and when available. Additionally, issues with digital literacy for both service providers and survivors necessitate training for providers and education/awareness for survivors on using digital platforms [60, 71, 100]. By addressing these challenges, the effectiveness and reach of virtual services can be significantly enhanced, ensuring that individuals can access the support they need.

Fig 4 visually illustrates the barriers, facilitators, and broader implications concerning research, interventions, and policies derived from a scoping review. It outlines significant issues such as the digital divide, weaponization of technology, and reduced personal

**Fig 4. Summary of barriers, facilitators and implications.**

connection in virtual interactions, emphasizing the need for innovative strategies and holistic supports. The figure also underscores the unique challenges posed by intersectionality, financial risks, and the essential nature of maintaining privacy, confidentiality, and safety. It advocates for robust policy changes, inclusive funding practices, and research that integrates the varied experiences of IPV Survivors during SLEs, aiming to enhance service accessibility and effectiveness.

## Strengths and limitations

One of the strengths of this study is that it provided an overview of the literature on access to and provision of services for IPV Survivors in the context of SLEs. This informed a larger study's methodology and content (i.e., interview tools and survey) which investigated the accessibility of IPV services during COVID-19 in Ontario, Canada. With a team of four reviewers, the scoping review offers a robust process to ensure articles that met the inclusion criteria were reviewed by at least two team members. Additionally, the first author screened and extracted all the papers, ensuring continuity throughout the review. Additionally, and with the support of a Librarian, an optimal combination of eight databases [141] was used, along with Google and Google Scholar to ensure comprehensive search strategy.

There are several limitations. First, over 90% of the articles found were related to COVID-19, with less than 10% related to economic recessions and natural/environmental disasters, therefore, there is an overwhelming representation (and bias) of the pandemic context in the findings. Second, one article abstract on economic recession was found but the full text was not available (and attempts to reach authors were not successful) for extraction and subsequent inclusion. Third, there is a bias in terms of perspective; many of the articles were from the formal service provider perspective, with less from the Survivor perspective, reinforcing the need for community-based research with the IPV community. Fourth, a quality appraisal was not completed as the main purpose of the review was to provide an overview of the literature rather than assess its quality. Fifth, the subjectivity may have arisen in study selection as authors could have included some articles which touched on but did not fully address accessibility to IPV services. Finally, an attempt was made to synthesize the findings and provide an overview of the facilitators, barriers and implications; however, they are heavily biased by the preponderance of recent COVID-19 events that accounted for a large part of the SLEs described in the literature as well as by the service provider perspectives.

## Conclusion

Our scoping review has mapped the barriers and facilitators affecting the accessibility of support services for Survivors of IPV during SLEs, including pandemics, natural disasters and economic recessions. The review found a critical need for all-encompassing and culturally sensitive support structures to mitigate the negative impact of SLEs on the health and well-being of IPV Survivors. This reinforces the vital necessity of continuous research and strategic emergency preparedness policy initiatives in the VAW domain, which is often overlooked, especially during SLEs.

## Supporting information

**S1 Checklist. PRISMA checklist.**
(PDF)

**S1 File. MEDLINE (OVID) search strategy.**
(PDF)

**S2 File. Extraction tool.**
(PDF)

**S1 Table. Supplement 3: Screening outcomes and reasons for inclusion/exclusion of articles in the scoping review (n = 5428 articles).**
(XLSX)

**S2 Table. Data extracted from eligible studies (n = 74).**
(XLSX)

**S3 Table. Barriers and Facilitators to accessing IPV supports during SLEs.**
(XLSX)

## Acknowledgments

The authors would like to acknowledge Karine Fournier, the Head of Reference Services at the University of Ottawa Library for her invaluable guidance while creating and finalizing the search strategy. We would like to acknowledge Julia Hajjar and Irfan Manji for proofreading and editing the manuscript.

## Author Contributions

**Conceptualization:** Dina Idriss-Wheeler, Sanni Yaya, Ziad El-Khatib.

**Data curation:** Dina Idriss-Wheeler, Xaand Bancroft, Saredo Bouraleh, Marie Buy.

**Formal analysis:** Dina Idriss-Wheeler, Xaand Bancroft, Saredo Bouraleh, Marie Buy.

**Investigation:** Dina Idriss-Wheeler.

**Methodology:** Dina Idriss-Wheeler.

**Project administration:** Dina Idriss-Wheeler.

**Resources:** Dina Idriss-Wheeler.

**Software:** Dina Idriss-Wheeler.

**Supervision:** Dina Idriss-Wheeler, Sanni Yaya, Ziad El-Khatib.

**Validation:** Dina Idriss-Wheeler.

**Writing – original draft:** Dina Idriss-Wheeler.

**Writing – review & editing:** Dina Idriss-Wheeler, Xaand Bancroft, Saredo Bouraleh, Marie Buy, Ziad El-Khatib.

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
