## [Decision Letter · Decision Letter 0]

14 Aug 2024

PONE-D-24-19008Exploring access to health and social supports for intimate partner violence (IPV) Survivors during stressful life events (SLEs) – A scoping reviewPLOS ONE

Dear Dr. El-Khatib,

Thank you for submitting your manuscript to PLOS ONE. After careful consideration, we feel that it has merit but does not fully meet PLOS ONE’s publication criteria as it currently stands. Therefore, we invite you to submit a revised version of the manuscript that addresses the points raised during the review process.

Below are the key points:

<ol><li>**Rationale and Methodology:**

Provide a clear rationale for focusing only on high-income countries.Include more detailed methodological considerations within the article, not just in the published protocol.Rationalize the selection of specific databases and the inclusion of individuals aged 13+.<li>**Inclusion/Exclusion Criteria:**

Avoid redundancy in criteria by ensuring exclusion criteria are not merely the opposite of inclusion criteria.Reframe and refine the eligibility criteria to improve clarity, particularly in Table 1.Justify the exclusion of articles focusing on low- and middle-income countries.<li>**Data Sources and Search Strategy:**

Specify the search terms used for grey literature and clarify the selection and screening process for these sources.Consider revising your PRISMA diagram to the 2020 version to better represent grey literature.<li>**Results and Discussion:**

Move detailed reporting of the number of sources from the methods section to the results.Discuss the inclusion of COVID-19-focused articles in light of existing literature on other stressful life events (SLEs).Consider the impact of focusing only on high-income countries on the findings and discuss articles excluded based on this criterion.<li>**Visual and Structural Improvements:**

Integrate Supplement 3 into the main body of the paper to enhance readability.Ensure consistency in punctuation, especially commas, and correct minor typographical errors. Please submit your revised manuscript by Sep 28 2024 11:59PM. If you will need more time than this to complete your revisions, please reply to this message or contact the journal office at plosone@plos.org. Please include the following items when submitting your revised manuscript:A rebuttal letter that responds to each point raised by the academic editor and reviewer(s). You should upload this letter as a separate file labeled 'Response to Reviewers'.A marked-up copy of your manuscript that highlights changes made to the original version. You should upload this as a separate file labeled 'Revised Manuscript with Track Changes'.An unmarked version of your revised paper without tracked changes. You should upload this as a separate file labeled 'Manuscript'.

We look forward to receiving your revised manuscript.

Kind regards,

Muhammad Shahzad Aslam, Ph.D.,M.Phil., Pharm-D

Academic Editor

PLOS ONE

Journal Requirements:

2. We note that this manuscript is a systematic review or meta-analysis; our author guidelines therefore require that you use PRISMA guidance to help improve reporting quality of this type of study. Please upload copies of the completed PRISMA checklist as Supporting Information with a file name “PRISMA checklist”.

Reviewers' comments:

Reviewer's Responses to Questions

**Comments to the Author**

1. Is the manuscript technically sound, and do the data support the conclusions?

Reviewer #1: Yes

Reviewer #2: Yes

2. Has the statistical analysis been performed appropriately and rigorously? 

Reviewer #1: N/A

Reviewer #2: N/A

3. Have the authors made all data underlying the findings in their manuscript fully available?

Reviewer #1: Yes

Reviewer #2: Yes

4. Is the manuscript presented in an intelligible fashion and written in standard English?

Reviewer #1: Yes

Reviewer #2: Yes

5. Review Comments to the Author

Reviewer #1: This is a very interesting, well written, clear and logical article. It provides results and an in-deep analysis with clear implications for practice and knowledge translation in different socio-structural levels in such a crucial topic as intimate partner violence in the context of stressful life events such as the emergence of COVID. This are some recommendations to strengthen/clarify certain points addressed in the article:

- Provide a clear rationale for focusing only on high-income countries.

- Authors mention that “The rationale for the inclusion and exclusion criteria is outlined in the published protocol, detailing the types of participants, key concepts, contexts, and evidence sources considered”. However, readers should be able to find all the methodological considerations in the article, at least a brief description.

- Consider that the contrary of an inclusion criterion it is not necessary an exclusion criterion, it is repetitive.

- Provide rational for the selection of those specific databases.

- Include in your analysis and discussion the SDG-2030, as the main agenda that allows us in present to discussion health related phenomena such as IPV as a complex and where multiple health determinants intervene (i.e. gender, geographical location, age etc.).

- IPV is a phenomenon crossed by gender as a critical and structural category, however, I consider that the review and its results need to be stated and discussed from this perspective in greater depth. For example, how is it that the national emergency plans (which do exist among the countries) do not consider gender as a category that differentiates health outcomes and whether, after COVID or other emergencies, it has been considered for inclusion.

Reviewer #2: Thank you for the opportunity to review this manuscript. The authors have done a commendable job in designing and conducting a robust scoping review, synthesizing a large body of literature clearly. This review will make a meaningful contribution to the literature.

Below are several specific suggestions to strengthen and clarify the manuscript. I would also encourage the authors to do a careful copy-edit to ensure consistent use of punctuation (particularly oxford commas) and catch the few instances of missing periods or odd spacing.

Methods

Protocol and Registration

• Please add a reference in for Tricco et al.

• Consider replacing the statement “Further details on the study can be found elsewhere (23).” And add the phrase “and the protocol has been peer reviewed and published (23).” To the end of the previous sentence.

Eligibility criteria

• The authors have a comprehensive description of inclusion criteria in Table 1, which is important for readers to understand the review. Readability/comprehension of this table would be improved with some streamlining.

o Consider refining each of the PICOT sections to just the element relevant to that section. From my understanding, that could be as follows.

population: individuals 13+ who have experienced IPV (include definition of IPV here)

• Articles focusing on service provider perspectives are included in this review, how are they accounted for in the eligibility criteria?

phenomenon of interest/context (intervention in original PICOT): stressful life events (include examples here)

high-income countries (specify what definitions/criteria were used here)

comparator: (none)

outcomes: access to formal and informal supports (include definitions of different types of supports here)

• Consider reframing the exclusion to positively identify articles that are ineligible, where possible (e.g., population younger than 13)

• This section would benefit from the rationale for the inclusion of individuals 13+, I appreciate this might be discussed in depth in the protocol, but the rationale should at least be mentioned here with the protocol referred to as relevant.

Information sources and search

• What search terms were used when searching for grey literature?

Selection of sources of evidence

• Consider removing the reporting on the number of records from this section and only reporting on it in the results section.

• How were grey literature sources selected and screened?

Data charting process

• Consider re-wording “automated extraction tool” – that description gives the impression that the extraction was conducted by the Covidence platform, rather than facilitated using their tool.

• Some clarification is needed for the sentence “Following consensus, the article was fully extracted and brought forward for the synthesis/analysis phase.”

• How were grey literature reports extracted?

o Was additional extraction completed after consensus was reached? Or was achieving consensus needed for the article to be considered fully extracted and moved to analysis?

Results

Synthesis of Sources of Evidence

• Consider moving the more fulsome reporting of number of reports from the methods section to this opening section of the results.

• Please also include reporting on the grey literature (both in the text and in the PRISMA diagram)

• I would strongly encourage the authors to update their PRISMA diagram to the revised 2020 version, which includes a pathway for reporting grey literature https://www.prisma-statement.org/prisma-2020-flow-diagram

Characteristics of sources of evidence

• Consider including the publication date range of the included articles. Given the high percentage of articles focusing on COVID-19, almost all will have been published in 2020 or later.

Barriers and Facilitators to accessing IPV services during Stressful Life Events (SLEs)

Overall the findings are presented well in a structure that is easy to follow, though long. Supplement 3 is an incredible visual summary of findings that would help make this section more easily digestible. I would highly encourage to editor to allow for inclusion of this visual (or an iteration of it) in the main body of the published work.

• In theme 5, the quote from Enarson should have the page number outside of the quotation marks.

Discussion

Context of Stressful Life Events (SLEs)

It seems odd that almost all the included articles focus on COVID-19 given that several reviews have been published in the last few years that have a much broader range of SLE covered (particularly related to climate change and natural disasters), some of which reported on service use and service provider experiences.

- E.g., Medzhitova et al. (2023) https://doi.org/10.1177/15248380221093688; van Daalen et al. 2022. doi:10.1016/s2542-5196(22)00088-2; Logie et al. 2024. doi:10.1080/17441692.2023.2299718

• Could the authors speak to how this body of literature intersects with the search results for this review? Were the articles noted in the above reviews captured in the search but excluded, or did the search not capture them?

A brief discussion of the 44 articles excluded because they were “not about access to services” would likely suffice here.

• It would also be interesting to discuss here, how the limitation to high income countries might impact the literature that was included (referring to the three article excluded for being about low and middle income countries).

Key Implications for Research, Interventions and Policy

• Figure 2 is a great summary of barriers, facilitators, and implications – it might be beneficial to present it earlier in the paper, perhaps adding a section to the end of the findings about implications and recommendations from the articles included in the review.

6. PLOS authors have the option to publish the peer review history of their article (what does this mean?). If published, this will include your full peer review and any attached files.

Reviewer #1: **Yes: **Alma Villa-Rueda

Reviewer #2: No

---

## [Author Response · Author response to Decision Letter 0]

23 Sep 2024

RESPONSE TO REVIEWERS

EDITOR’S COMMENTS

Below are the key points:

1. Rationale and Methodology:

o Provide a clear rationale for focusing only on high-income countries.

RESPONSE: Thank you. The section has been revised accordingly with a clear rationale for focusing on HICs.

o Include more detailed methodological considerations within the article, not just in the published protocol.

RESPONSE: Thank you. More detailed methodological considerations have been added as requested.

o Rationalize the selection of specific databases and the inclusion of individuals aged 13+.

RESPONSE: Thank you. Rationale and explanation for the specific databases and inclusion of individuals ages 13+ has been added.

2. Inclusion/Exclusion Criteria:

o Avoid redundancy in criteria by ensuring exclusion criteria are not merely the opposite of inclusion criteria.

o Reframe and refine the eligibility criteria to improve clarity, particularly in Table 1.

o Justify the exclusion of articles focusing on low- and middle-income countries.

RESPONSE: Thank you. The entire section has been revised accordingly for a clear concise eligibility criterion along with justification of articles focusing on HIC (as indicated above).

3. Data Sources and Search Strategy:

o Specify the search terms used for grey literature and clarify the selection and screening process for these sources.

o Consider revising your PRISMA diagram to the 2020 version to better represent grey literature.

RESPONSE: Thank you for bringing this to our attention. In our initial manuscript submission, we mentioned plans to include grey literature in our review and this was erroneously transferred to our current manuscript. After searching 8 databases and due to an overwhelming number of peer-reviewed articles that met our inclusion criteria, we ultimately decided against incorporating grey literature and sticking to peer-reviewed articles. We apologize for any confusion this may have caused and have taken out the grey literature statement from our submission which was erroneously included. We have provided an explanation of the deviation from the protocol

4. Results and Discussion:

o Move detailed reporting of the number of sources from the methods section to the results.

RESPONSE: Thank you. Revised as suggested. 

o Discuss the inclusion of COVID-19-focused articles in light of existing literature on other stressful life events (SLEs).

RESPONSE: Thank you. Revised as suggested. Please refer to the section in Reviewer #2.

o Consider the impact of focusing only on high-income countries on the findings and discuss articles excluded based on this criterion.

RESPONSE: Thank you. Please note that we have provided a rationale for why we focus on HICs. This has been described in detail for both Reviewers below. 

5. Visual and Structural Improvements:

o Integrate Supplement 3 into the main body of the paper to enhance readability.

RESPONSE: Thank you, we have integrated a version of supplement 3 into the body (as Figures 3 and 4). The actual supplement is too large to include so an iteration (as suggested by reviewer #2) has been included.

o Ensure consistency in punctuation, especially commas, and correct minor typographical errors.

RESPONSE: Thank you, completed.

RESPONSE: The style requirements have been considered and incorporated into the revised version.

7. We note that this manuscript is a systematic review or meta-analysis; our author guidelines therefore require that you use PRISMA guidance to help improve reporting quality of this type of study. Please upload copies of the completed PRISMA checklist as Supporting Information with a file name “PRISMA checklist”.

RESPONSE: Thank you, this was completed and is submitted as part of the package, entitled PRISMA Fillable Checklist.

8. Please include a separate caption for each figure in your manuscript.

RESPONSE: We have included separate captions for each figure.

RESPONSE: We have included captions for the Supporting Information at the end of the manuscript and updated in-text citations to match accordingly.

REVIEWER #1 COMMENTS

This is a very interesting, well written, clear and logical article. It provides results and an in-deep analysis with clear implications for practice and knowledge translation in different socio-structural levels in such a crucial topic as intimate partner violence in the context of stressful life events such as the emergence of COVID. 

RESPONSE: Thank you for your thoughtful and kind comment. 

Provide a clear rationale for focusing only on high-income countries. 

RESPONSE: Thank you. We have added the rationale for focusing only on high-income countries [page 6, lines 103-108]. In summary, the focus is on high-income countries because the scoping review is part of (and to inform) a larger study looking at access to both informal (family, friends, neighbours) and formal violence against women (VAW) supports for individuals who experienced IPV during COVID-19 lockdowns in Canada.

Authors mention that “The rationale for the inclusion and exclusion criteria is outlined in the published protocol, detailing the types of participants, key concepts, contexts, and evidence sources considered”. However, readers should be able to find all the methodological considerations in the article, at least a brief description. Consider that the contrary of an inclusion criterion it is not necessary an exclusion criterion, it is repetitive.

RESPONSE: Thank you and we agree. The entire Eligibility section has been reorganized and for better readability. Reviewer #2 also provided similar feedback and kindly suggested an approach which we have now incorporated [p. 8-9, lines 133-173].

Provide rational for the selection of those specific databases.

RESPONSE: Thank you. The rationale for selection of specific databases was included. [p. 10, lines 181-184].

Include in your analysis and discussion the SDG-2030, as the main agenda that allows us in present to discussion health related phenomena such as IPV as a complex and where multiple health determinants intervene (i.e. gender, geographical location, age etc.).

RESPONSE: Thank you for your suggestion of incorporating the discussion on SDG-2030 main agenda in the context of IPV. We have added a section in the human rights social justice lens section of the discussion. [p. 60, lines 982-997).

IPV is a phenomenon crossed by gender as a critical and structural category, however, I consider that the review and its results need to be stated and discussed from this perspective in greater depth. For example, how is it that the national emergency plans (which do exist among the countries) do not consider gender as a category that differentiates health outcomes and whether, after COVID or other emergencies, it has been considered for inclusion.

RESPONSE: Thank you for your suggestion. We have incorporated a section in the discussion regarding gender as a critical category in emergency planning and management. [p. 58-59, lines 953-964]

REVIEWER #2 COMMENTS

Thank you for the opportunity to review this manuscript. The authors have done a commendable job in designing and conducting a robust scoping review, synthesizing a large body of literature clearly. This review will make a meaningful contribution to the literature.

RESPONSE: Thank you very much for your kind words and comments. 

Below are several specific suggestions to strengthen and clarify the manuscript. I would also encourage the authors to do a careful copy-edit to ensure consistent use of punctuation (particularly oxford commas) and catch the few instances of missing periods or odd spacing.

RESPONSE: Thank you very much. We have reviewed the entire manuscript for consistency in punctuation and missing periods or odd spacing. 

Methods > Protocol and Registration > Please add a reference in for Tricco et al. 

RESPONSE: Thank you for catching that; we have now added the reference for Tricco et al. (2018). [p. 6, line 125]

Consider replacing the statement “Further details on the study can be found elsewhere (23).” And add the phrase “and the protocol has been peer reviewed and published (23).” To the end of the previous sentence.

RESPONSE: Thank you. We have revised the sentence as suggested. [p. 7, lines 126-127]

Eligibility criteria

The authors have a comprehensive description of inclusion criteria in Table 1, which is important for readers to understand the review. Readability/comprehension of this table would be improved with some streamlining.

o Consider refining each of the PICOT sections to just the element relevant to that section. From my understanding, that could be as follows.

population: individuals 13+ who have experienced IPV (include definition of IPV here)

• Articles focusing on service provider perspectives are included in this review, how are they accounted for in the eligibility criteria?

phenomenon of interest/context (intervention in original PICOT): stressful life events (include examples here)

high-income countries (specify what definitions/criteria were used here)

comparator: (none)

outcomes: access to formal and informal supports (include definitions of different types of supports here)

Consider reframing the exclusion to positively identify articles that are ineligible, where possible (e.g., population younger than 13). This section would benefit from the rationale for the inclusion of individuals 13+, I appreciate this might be discussed in depth in the protocol, but the rationale should at least be mentioned here with the protocol referred to as relevant.

RESPONSE: RESPONSE: Thank you very much for your comments and wonderful suggestion to organize the eligibility criteria section. Reviewer #1 also provided a similar comment regarding this section. We have revised it accordingly, using your suggestion of PICOR. We have reframed the exclusion criteria and added the rationale for why individuals 13+ were included (p. 8-9, lines 133-173). 

Information sources and search

• What search terms were used when searching for grey literature?

RESPONSE: Thank you for bringing this to our attention. In our initial manuscript submission, we mentioned plans to include grey literature in our review. However, after searching 8 databases and due to an overwhelming number of peer-reviewed articles that met our inclusion criteria, we ultimately decided against incorporating grey literature and sticking to peer-reviewed articles. We apologize for any confusion this may have caused and have taken out the grey literature statement from our submission which was erroneously included. We have provided an explanation of the deviation from the protocol [P. 10; lines 184-189].

Selection of sources of evidence

• Consider removing the reporting on the number of records from this section and only reporting on it in the results section.

RESPONSE: Thank you. We have removed the reporting on the number of records from this section and only reporting on it in the results section. [p.10, lines 192-197].

• How were grey literature sources selected and screened?

RESPONSE: Please see above explanation. 

Data charting process

• Consider re-wording “automated extraction tool” – that description gives the impression that the extraction was conducted by the Covidence platform, rather than facilitated using their tool. • Some clarification is needed for the sentence “Following consensus, the article was fully extracted and brought forward for the synthesis/analysis phase.” Was additional extraction completed after consensus was reached? Or was achieving consensus needed for the article to be considered fully extracted and moved to analysis?

RESPONSE: Thank you and we agree. Achieving consensus by the two extractors was needed for the article to be considered fully extracted and move to analysis phase. We have revised the section accordingly. [p. 11, lines 216-220]

• How were grey literature reports extracted?

RESPONSE: Please see above explanation regarding the grey literature. 

Results

Synthesis of Sources of Evidence

• Consider moving the more fulsome reporting of number of reports from the methods section to this opening section of the results.

RESPONSE: Thank you. We have revised the section accordingly, the fulsome reporting of numbers were moved from the methods to the results section. [p. 12, lines 240-244].

• Please also include reporting on the grey literature (both in the text and in the PRISMA diagram).

• I would strongly encourage the authors to update their PRISMA diagram to the revised 2020 version, which includes a pathway for reporting grey literature https://www.prisma-statement.org/prisma-2020-flow-diagram

RESPONSE: Thank you very much for the new source and bringing this to our attention. We have updated our PRISMA Diagram as suggested. 

Characteristics of sources of evidence

• Consider including the publication date range of the included articles. Given the high percentage of articles focusing on COVID-19, almost all will have been published in 2020 or later.

RESPONSE: Great suggestion, thank you. We have added the sentence on publication date range. [p. 13, lines 252-253]

Barriers and Facilitators to accessing IPV services during Stressful Life Events (SLEs)

Overall the findings are presented well in a structure that is easy to follow, though long. Supplement 3 is an incredible visual summary of findings that would help make this section more easily digestible. I would highly encourage to editor to allow for inclusion of this visual (or an iteration of it) in the main body of the published work.

RESPONSE: Thank you. An iteration of supplement 3 has been included in the body of the paper. 

• In theme 5, the quote from Enarson should have the page number outside of the quotation marks.

RESPONSE: Thank you very much. Revised as suggested. [p. 43, line 582]

Discussion

Context of Stressful Life Events (SLEs)

It seems odd that almost all the included articles focus on COVID-19 given that several reviews have been published in the last few years that have a much broader range of SLE covered (particularly related to climate change and natural disasters), some of which reported on service use and service provider experiences.

- E.g., Medzhitova et al. (2023) https://doi.org/10.1177/15248380221093688; van Daalen et al. 2022. doi:10.1016/s2542-5196(22)00088-2; Logie et al. 2024. doi:10.1080/17441692.2023.2299718

Could the authors speak to how this body of literature intersects with the search results for this review? Were the articles noted in the above reviews captured in the search but excluded, or did the search not capture them? A brief discussion of the 44 articles excluded because they were “not about access to services” would likely suffice here.

RESPONSE: Thank you for your valuable comments. This scoping review specifically focused on accessing or providing formal and informal services from the perspectives of IPV survivors and support providers during Stressful Life Events (SLEs). Our review was not intended to examine the prevalence, incidence, likelihood, or associations between SLEs and IPV or GBV, which explains some of the differences in article inclusion compared to other reviews mentioned.

Regarding the specific reviews you noted:

- Medzhitova et al. (2023): Among the 24 studies reviewed, 14 were conducted in high-income countries (HICs) and 10 in low- and middle-income countries (LMICs). We excluded the studies from LMICs for reasons discussed in our limitations section. Of the remaining 14 articles from HICs, only 2 discussed access to services during SLEs. The m

---

## [Decision Letter · Decision Letter 1]

29 Oct 2024

Exploring access to health and social supports for intimate partner violence (IPV) Survivors during stressful life events (SLEs) – a scoping review

PONE-D-24-19008R1

Dear Dr. El-Khatib,

We’re pleased to inform you that your manuscript has been judged scientifically suitable for publication and will be formally accepted for publication once it meets all outstanding technical requirements.

Kind regards,

Muhammad Shahzad Aslam, Ph.D.,M.Phil., Pharm-D

Academic Editor

PLOS ONE

Additional Editor Comments (optional):

Reviewers' comments:

Reviewer's Responses to Questions

**Comments to the Author**

1. If the authors have adequately addressed your comments raised in a previous round of review and you feel that this manuscript is now acceptable for publication, you may indicate that here to bypass the “Comments to the Author” section, enter your conflict of interest statement in the “Confidential to Editor” section, and submit your "Accept" recommendation.

Reviewer #1: All comments have been addressed

2. Is the manuscript technically sound, and do the data support the conclusions?

Reviewer #1: Yes

3. Has the statistical analysis been performed appropriately and rigorously? 

Reviewer #1: Yes

4. Have the authors made all data underlying the findings in their manuscript fully available?

Reviewer #1: Yes

5. Is the manuscript presented in an intelligible fashion and written in standard English?

Reviewer #1: Yes

6. Review Comments to the Author

Reviewer #1: Authors have addressed all the comments in a clear and logic way. Now the article is stated from a specific geographic and temporal space, framed in the current public policy agenda. It adds depth to the gender category.

7. PLOS authors have the option to publish the peer review history of their article (what does this mean?). If published, this will include your full peer review and any attached files.

Reviewer #1: **Yes: **ALMA ANGELICA VILLA RUEDA

---

## [Editor Report · Acceptance letter]

17 Nov 2024

PONE-D-24-19008R1 

PLOS ONE

Dear Dr. El-Khatib, 

I'm pleased to inform you that your manuscript has been deemed suitable for publication in PLOS ONE. Congratulations! Your manuscript is now being handed over to our production team.

Kind regards, 

on behalf of

Dr. Muhammad Shahzad Aslam 

Academic Editor

PLOS ONE